# Saturation genome editing of DDX3X clarifies pathogenicity of germline and somatic variation

Elizabeth J. Radford [1,2,8], Hong-Kee Tan[1,8], Malin H. L. Andersson [1], James D. Stephenson[3], Eugene J. Gardner [4], Holly Ironfield[1], Andrew J. Waters[1], Daniel Gitterman[1], Sarah Lindsay [1], Federico Abascal [1], Iñigo Martincorena [1], Anna Kolesnik-Taylor[5], Elise Ng-Cordell[5,6], Helen V. Firth[1,7], Kate Baker[5,7], John R. B. Perry [4], David J. Adams [1], Sebastian S. Gerety [1,9] & Matthew E. Hurles [1,9] ✉

Loss-of-function of *DDX3X* is a leading cause of neurodevelopmental disorders (NDD) in females. *DDX3X* is also a somatically mutated cancer driver gene proposed to have tumour promoting and suppressing effects. We perform saturation genome editing of *DDX3X*, testing in vitro the functional impact of 12,776 nucleotide variants. We identify 3432 functionally abnormal variants, in three distinct classes. We train a machine learning classifier to identify functionally abnormal variants of NDD-relevance. This classifier has at least 97% sensitivity and 99% specificity to detect variants pathogenic for NDD, substantially out-performing in silico predictors, and resolving up to 93% of variants of uncertain significance. Moreover, functionally-abnormal variants can account for almost all of the excess nonsynonymous *DDX3X* somatic mutations seen in *DDX3X*-driven cancers. Systematic maps of variant effects generated in experimentally tractable cell types have the potential to transform clinical interpretation of both germline and somatic disease-associated variation.

The disease relevance of the vast majority of variants in disease-associated genes is not known and is challenging to predict. In diagnostic practice, genetic variants are commonly classified according to the American College of Medical Genetics (ACMG/AMP) guidelines, combining information across different types of evidence[1,2]. Variants with insufficient or conflicting evidence of their effect are classified as variants of uncertain significance (VUS). With increased diagnostic use of genomic sequencing, the number of VUS has grown rapidly[3]. Conventional strategies to resolve VUS rely on the accumulation of clinical and population data, the use of in silico predictors of variant effect or small-scale retrospective functional studies.

Multiplexed assays of variant effect (MAVEs) that prospectively assess variant effects systematically have numerous advantages over small-scale retrospective functional studies[3]. Saturation genome editing (SGE) is a MAVE technique utilising CRISPR/Cas9-stimulated homology-directed repair (HDR) to introduce specified genetic variants into an endogenous locus. Importantly, because genomic and regulatory context is preserved, non-coding and synonymous variants

[1]Wellcome Sanger Institute, Hinxton CB10 1SA, UK. [2]Department of Paediatrics, University of Cambridge, Level 8, Cambridge Biomedical Campus, Cambridge CB2 0QQ, UK. [3]EMBL-EBI, Wellcome Genome Campus, Hinxton CB10 1SD, UK. [4]MRC Epidemiology Unit, University of Cambridge School of Clinical Medicine, Cambridge Biomedical Campus, Cambridge CB2 0QQ, UK. [5]MRC Cognition and Brain Sciences Unit, University of Cambridge, Cambridge, UK. [6]Department of Psychology, University of British Columbia, Vancouver, Canada. [7]Department of Medical Genetics, University of Cambridge, Cambridge, UK. [8]These authors contributed equally: Elizabeth J. Radford, Hong-Kee Tan. [9]These authors jointly supervised this work: Sebastian S. Gerety, Matthew E. Hurles. ✉e-mail: meh@sanger.ac.uk

that alter splicing or transcriptional control can also be assessed[4,5]. SGE has been demonstrated to effectively resolve most VUS in *BRCA1*[6], outperforms in silico prediction algorithms[6], and has proven diagnostic utility[5,7].

Pathogenic variation in *DDX3X*, an X-linked gene encoding an RNA helicase, has been robustly associated with both neurodevelopmental disorders (NDD) and cancer (somatic mutations). Heterozygous *DDX3X* loss-of-function variants are one of the most common genetic causes of intellectual disability in females[8,9], with an approximately equal split between protein-truncating and missense variants. However, variable and non-specific presentation hampers accurate variant interpretation. Protein-truncating variants are not seen in males, presumably because they are hemizygous lethal[10]. It has been proposed that some *DDX3X* missense variants in males can also cause NDD[11], however, these are observed much less frequently than in females and the statistical evidence for mutation enrichment in NDD-affected individuals is not conclusive[12]. *DDX3X* has also been identified as a somatic driver gene in several different cancers, with most compelling evidence in medulloblastoma[13–15]. Uncertainty persists as to whether *DDX3X* acts as an oncogene or tumour suppressor[16,17]. *DDX3X* was identified as an essential gene in a genome-wide CRISPR essentiality screen performed in the HAP1 cell line[18,19], suggesting that *DDX3X* might be suitable for SGE in HAP1 cells.

Here we characterise the functional consequence of over 12,000 *DDX3X* variants, using an improved SGE-based assay, to assess their relevance for NDDs and cancer. We demonstrate the utility of these data for clinical variant classification in NDDs, and demonstrate that *DDX3X* predominantly acts as a tumour suppressor gene across multiple tumour types.

## Results

### Saturation genome editing of *DDX3X*

We performed SGE on all 17 coding exons of *DDX3X*, with two independent sgRNAs and HDR template libraries per exon (Fig. 1a), performed in triplicate over a time course of 21 days (Fig. 1b) with cells harvested at five timepoints (Days 4, 7, 11, 15 and 21 post-transfection).

Collectively, the HDR template libraries contained all possible 6156 coding single nucleotide variants (SNVs); all 626 deletions of contiguous codons; all 51 insertions and deletions less than 50 bp reported in ClinVar, DECIPHER, GnomAD or UK Biobank; and 2247 non-coding SNVs and 2 bp deletions located within 25 bp 5′ and 15 bp 3′ of the exon boundary. In addition, for each coding SNV, at least one 'redundant variant' (typically a multinucleotide variant) was designed (3696 variants), to assess the concordance of the effect of the same amino-acid change generated by two different DNA changes.

For each variant, we calculated the log2 fold-change (LFC) for each guide at each timepoint, relative to the median of the variants expected to have no functional impact (synonymous and intronic variants). We also calculated a single functional score (LFC-trend) for each variant across timepoints, corresponding to the rate of change in relative abundance over time. The two independent sgRNA experiments within each exon were highly correlated for both measures of variant abundance (Pearson's correlation: LFC-trend = 0.897, LFC: 0.886–0.913 for each timepoint; Fig. S1a–e). Measures of variant abundance for the same variant across the two sgRNA experiments for each exon were combined using a weighted mean. The resultant combined LFC-trend score (cLFC-trend) is highly correlated with the combined-LFC (cLFC) of variant abundance at Day 15 (Pearson $r$ = 0.995, Fig. S1f). The statistical significance of the cLFC of a variant differing from that of synonymous and intronic variants within the same exon was quantified using the Benjamini-Hochberg (BH) corrected False Discovery Rate (FDR).

We identified 2337 significantly (Day 15 cLFC FDR ≤ 0.01) SGE-depleted variants and 1095 significantly (Day 15 cLFC FDR ≤ 0.01) SGE-enriched variants. Downstream analyses focussed on SNVs, excluding

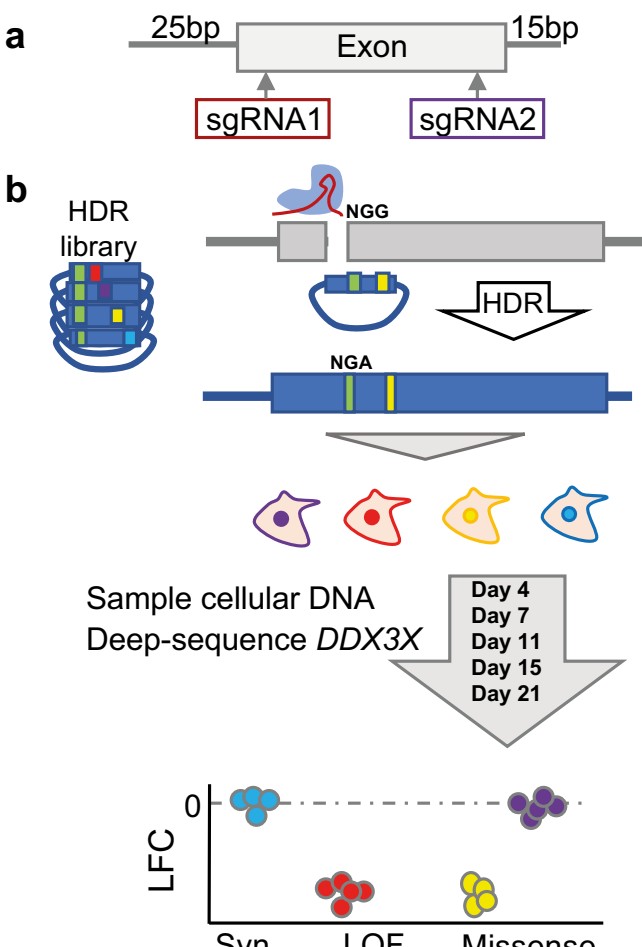

**Fig. 1 | Experimental design overview. a** Two independent sgRNAs and associated HDR variant libraries are designed at the 5′ and 3′ end of each exon. **b** The sgRNA, together with the HDR template library are transfected into *LIG4*-KO Cas9-expressing HAP1 cells. HDR utilises the library as a template for repair of the sgRNA-directed double-stranded DNA cut, incorporating a *DDX3X* variant of interest. Damaging *DDX3X* variants will reduce cell viability or proliferation. Variant abundance was assessed at five timepoints. Functional missense (purple) and synonymous variants (Syn, blue) remain abundant, while loss-of-function variants (LOF, red), and damaging missense (yellow) variants are depleted.

redundant multinucleotide variants. Variants with cLFC FDR > 0.01 on Day 15 were classified as "SGE-unchanged". Missense variants were very significantly over-represented among SGE-enriched variants, by 1.5-fold ($X^2$ $p$ = 3.2 × 10⁻³⁶), suggesting that they are unlikely to represent technical artefacts. We analysed the kinetics of variant depletion using a two-dimensional Gaussian mixture model of Day 7 and Day 15 cLFCs (Figs. 2 and S2a–d). This identified two classes of depleted variants: slow-depleting and fast-depleting. The map of all variants' functional class across the gene is shown in Fig. S3.

### Structural features of depleted and enriched variants

Both amino acid conservation[20] and CADD scores[21] are significantly higher for SGE-enriched and SGE-depleted SNVs compared to SGE-unchanged SNVs (Dunn's post-test FDRs < 1 × 10⁻²³, see Fig. 3a). Compared to SGE-unchanged variants, SGE-depleted missense and codon deletion variants are more likely to be found deep within the DDX3X protein (Figs. 3a–d and S2e), with fast-depleting variants closer to the centroid of the protein than slow-depleting variants (Dunn's post-test missense FDR = 4.8 × 10⁻⁶, codon-deletion FDR = 4.8 × 10⁻⁵, see Figs. 3a–d and S2e). In addition, compared to

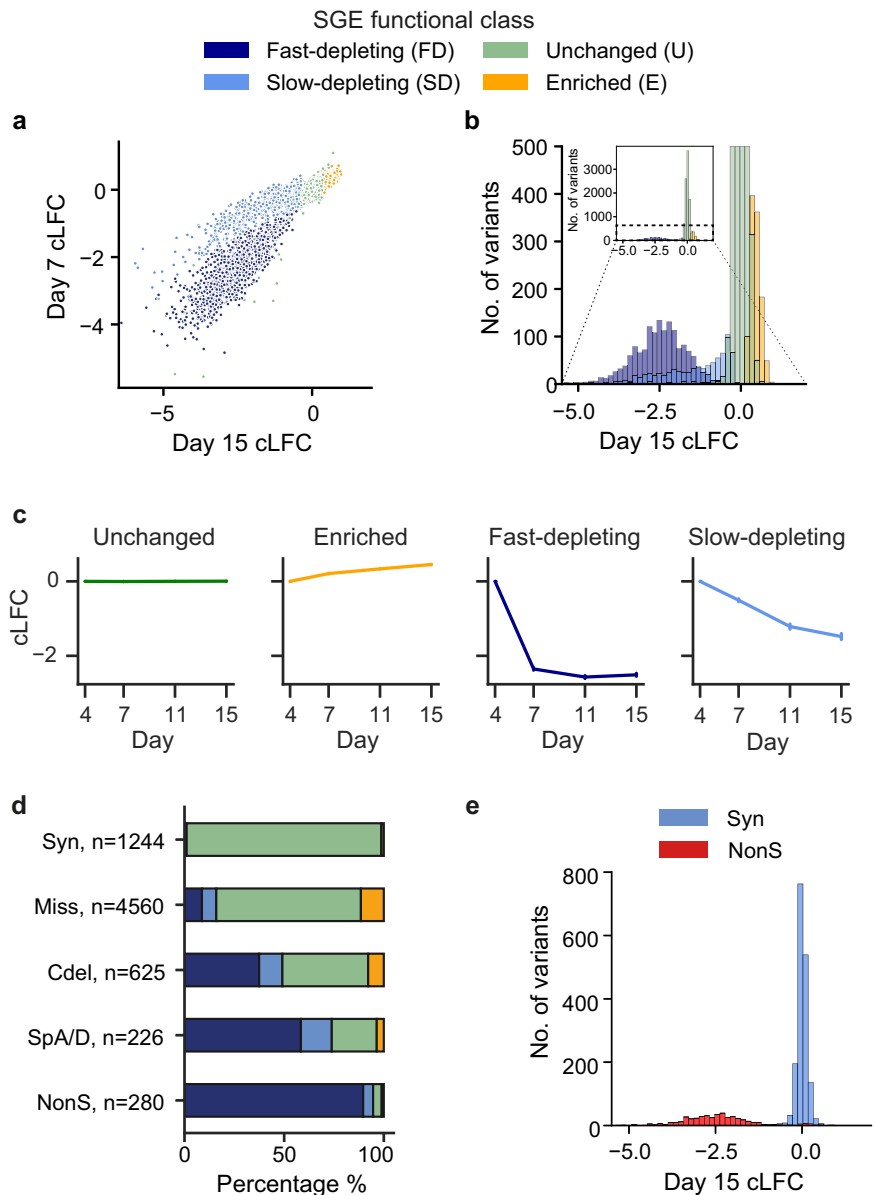

**Fig. 2 | Functional classification of *DDX3X* variants. a** Day 7 and Day 15 cLFC of variant abundance. **b** Day 15 cLFC of variant abundance. **c** Average cLFC for all variants in each SGE functional class for each time point. Number of variants per class: unchanged *n* = 9344; enriched = 1095, fast-depleting = 1546, slow-depleting = 791. Error bars represent the 95% CI. **d** Proportions of SGE functional classes within the distribution of all missense and codon-deletion variants, single nucleotide variant synonymous (Syn), missense (Miss), codon deletion (Cdel), canonical splice acceptor/donor (SpA/D) and nonsense (NonS) variants. **e** Day 15 cLFC of variant abundance for synonymous (*n* = 1244) and nonsense (*n* = 280) variants only. Source data are provided as a Source Data file.

SGE-depleted, particularly fast-depleting variants are over-represented within the helicase domains and at residues that interact with RNA, ATP and $Mg^{2+}$ ($\chi^2$ *p* < 0.001, see Fig. 3b–d). Conversely, SGE-enriched missense and codon-deletion variants are significantly under-represented within the helicase domains ($\chi^2$ *p* < 0.01, see Fig S2e, f and Supplementary movies 1 and 2). SGE-enriched, fast and slow-depleting missense variants are more disruptive to protein structure (as predicted by the change in Gibbs free energy of folding ($\Delta\Delta$G))[22] than SGE-unchanged missense variants (Fig. 3a, Dunn's post-test FDRs < 1 × 10$^{-23}$). There is no significant difference between the $\Delta\Delta$G of fast and slow-depleting variants (Fig. 3a, Dunn's post-test FDR = 0.136).

Both SGE-depleted and SGE-enriched variants are observed less frequently than expected under a germline mutational model[23] in the Genome Aggregation Database (GnomAD) and UK Biobank (UKBB)

(Fig. 3a), consistent with protein-damaging *DDX3X* variation being underrepresented in healthy population cohorts due to negative selection. Although SGE-enriched variants are not selected against to as great an extent as SGE-depleted variants, a phenome-wide association analysis of SGE-enriched variants in UKBB identified no statistically significant associations (Supplementary data S1), but had low statistical power due to the paucity of individuals (*N* = 35) carrying these variants.

## Functional classification of synonymous, non-synonymous and intronic variants

We observed that 98% (1222/1244) of synonymous SNVs were SGE-unchanged (Figs. 2d, e and S3–5). The 11 SGE-depleted synonymous SNVs include two that are predicted to have a negative impact on splicing by SpliceAI[24]. Both were predicted to create novel splice-donors.

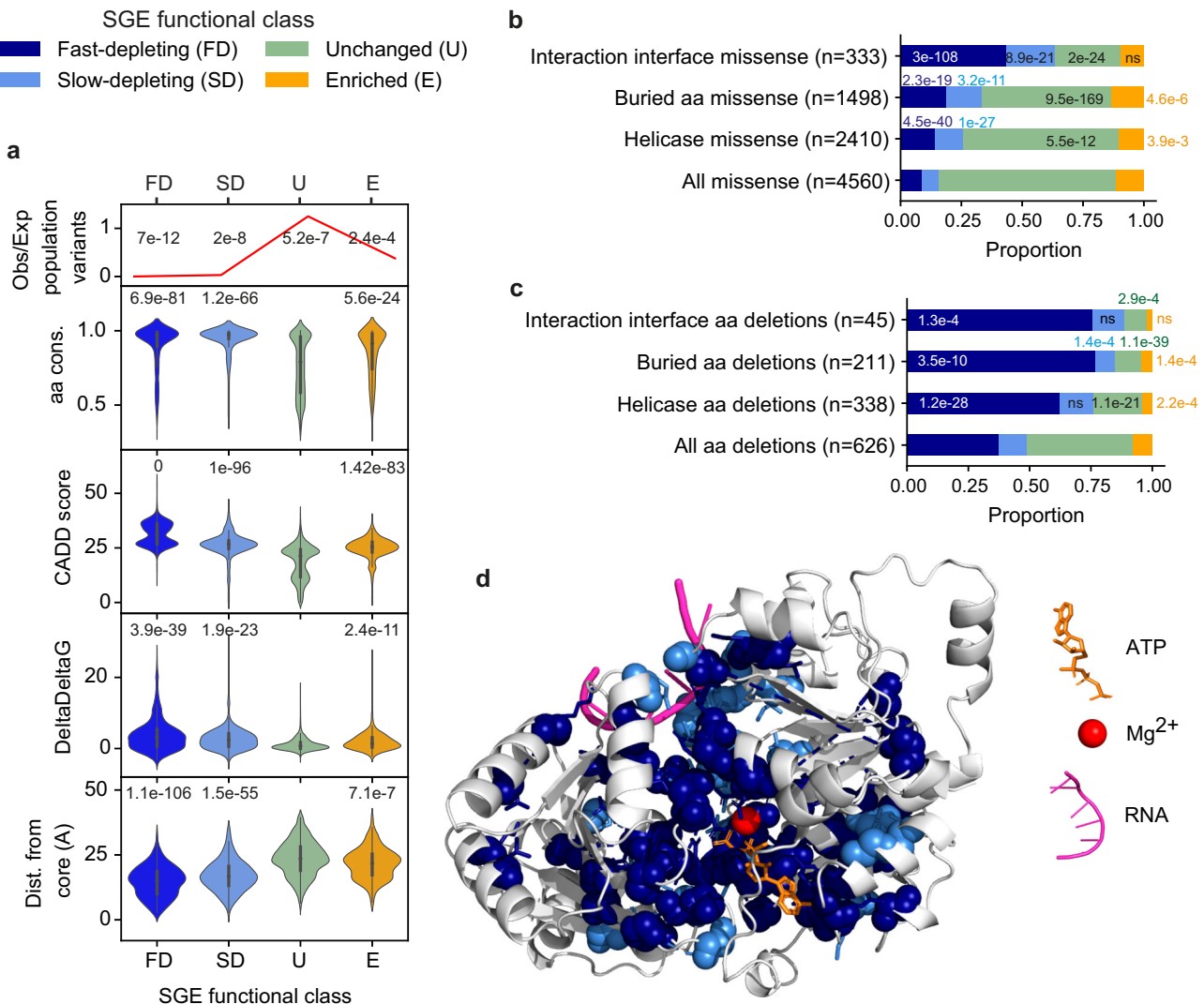

**Fig. 3 | Properties of SGE-depleted and SGE-enriched variants. a** For each SGE functional class: Top panel: Observed/Expected number of DDX3X SNVs in UK Biobank (UKBB) and Genome Aggregation Database (GnomAD), number of variants per class: unchanged $n$ = 6732; enriched = 710, fast-depleting = 1108, slow-depleting = 710. $X^2$ test, degrees of freedom (df) = 3. Second panel: Amino acid conservation, Kruskal–Wallis (KW) test $p = 1.1 \times 10^{-141}$. Third panel: CADD PHRED scores, KW test $p$ - 0. Fourth panel: ΔΔG for missense variants, KW test $p = 1.8 \times 10^{-54}$. Lower panel: distance from the centroid of DDX3X to the amino-acid side chain centroid (ångströms), missense variants, KW test $p = 2.0 \times 10^{-234}$. Dunn's post-test FDR, corrected for multiple testing by the Benjamini–Hochberg (BH) method, is shown in panels 2–5. Internal boxplots within each violinplot show median and interquartile range (IQR), whiskers denote 1.5 x IQR. **b** The proportion of SGE functional classes in *DDX3X* missense variants stratified by their position in the protein. Interaction interface: residues in contact with RNA, magnesium ion or ATP. Buried residues: all residues with total solvent accessible surface area <25%. $X^2$ $p$-values relative to all missense are shown, df = 3. **c** The proportion of SGE functional classes in DDX3X codon-deletion variants stratified by their position in the protein. $X^2$ $p$-values relative to all codon deletions are shown, df = 3. **d** AlphaFold2 DDX3X structure together with ATP, magnesium ion and RNA. Coloured according to the modal SGE functional class for missense variants at each residue. Spheres: residue main chain. Sticks: residue side chain. Source data are provided as a Source Data file.

We found that 95% (265/280) of nonsense SNVs were SGE-depleted, including 99.5% (219/220) of those predicted to trigger nonsense-mediated decay (NMD)[25] (Figs. 2d, e, 4a, bi, bii and S3–5). The majority (95%) of SGE-depleted nonsense variants were fast-depleting. 7/21 nonsense variants predicted to escape NMD at the 3′ end of the gene were in fact SGE-depleted, 2 fast and 5 slow-depleting (Figs. 2d and 4a, bi, bii). This suggests that some nonsense variants predicted in silico to escape NMD may still have a negative functional effect.

Sixteen percent (727/4560) of missense variants are SGE-depleted, 55% of which are fast-depleting (Fig. 2d and 4a–c). 49% (307/625) of inframe codon-deleting variants are SGE-depleted, 76% of which are fast-depleting. SGE-depleted missense and codon-deletion variants are strongly enriched within previously defined protein domains of DDX3X (Figs. 3b, c, 4a–c and S3–5; missense and codon-deletion variants: $X^2$ $p$-value 2.1 × 10⁻⁶⁶, 7.7 × 10⁻²⁷, respectively) with 92% of all SGE-depleted missense variants occurring within the helicase domains and helicase Q motif. In exon 11, 90% (26/29) of missense variants in the DEAD box domain are SGE-depleted (Fig. 4a). In exon 15, we observe an abrupt change in the functional impact of missense and codon-deletion variants which coincides precisely with the UniProt-predicted end of the C-terminal helicase domain (Fig. 4a). Throughout the gene, 11.5% (524/4560) of missense variants and 16% (49/625) of codon-deletion variants are SGE-enriched.

At canonical splice acceptor and donor sites, 74% (167/226) of variants are SGE-depleted, a lower proportion than for nonsense variants, 3.5% (8/226) SGE-enriched and 23% (51/226) unchanged. Canonical splice site variants that are SGE-unchanged have significantly lower SpliceAI scores than variants that are fast-depleting (Fig. 4d, Kruskal–Wallis $p$ = 0.002, Dunn's BH-corrected FDR = 0.001). Exon 3 has a strikingly high proportion, 66% (10/15), of SGE-unchanged

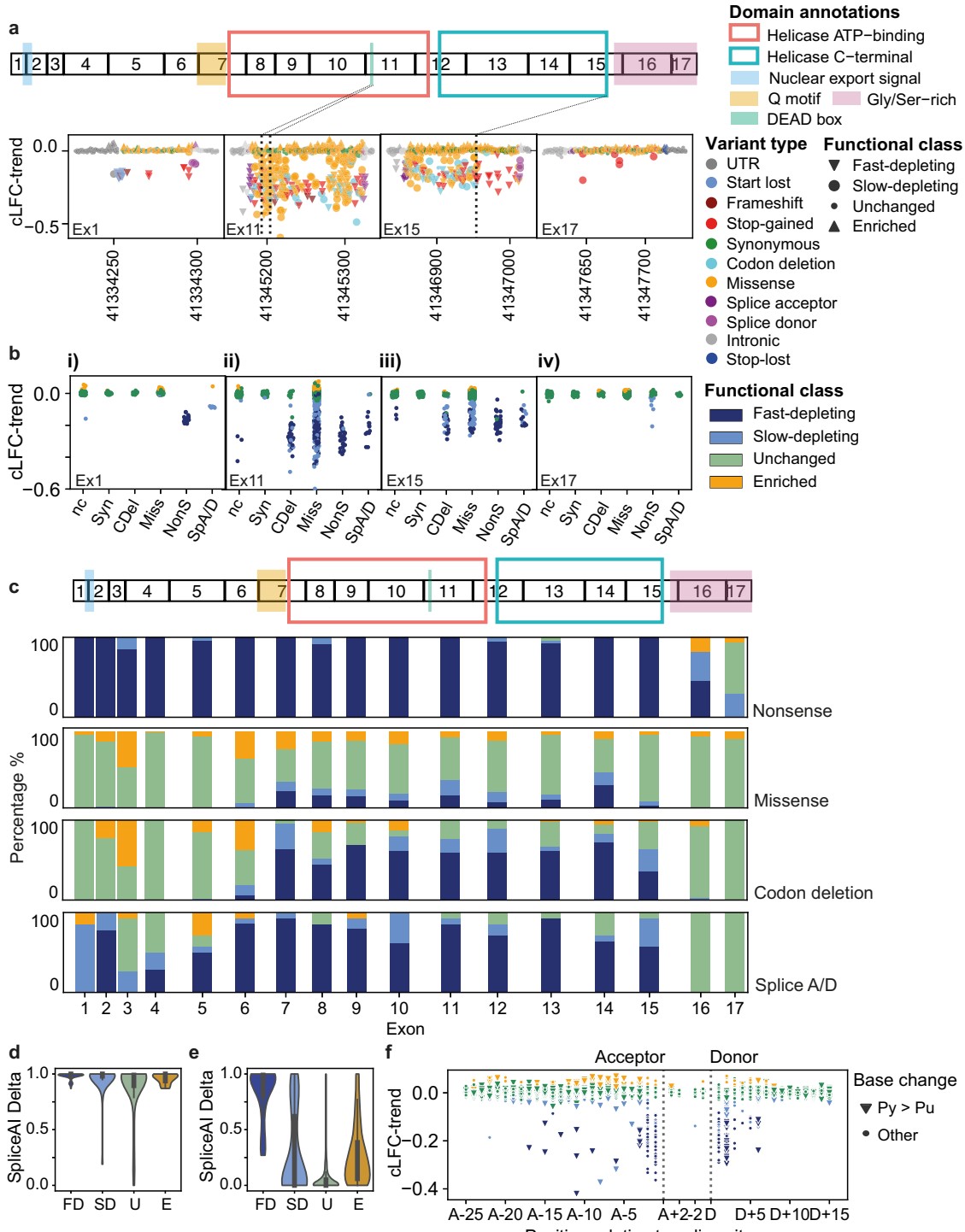

**Fig. 4 | Functional characterisation of *DDX3X* variant types. a** Top panel: *DDX3X* exon structure with locations of key domains and protein annotations. Lower panel: cLFC-trend plotted against chromosome coordinate for variants in four exons. **b** cLFC-trend of non-coding (nc), synonymous (Syn), in-frame codon-deletion (Cdel), missense (Miss), nonsense (NonS) and canonical splice acceptor/donor SNVs (SpA/D) in the same four exons, coloured by SGE functional class. **c** Proportion of SGE functional classes across *DDX3X* exons for nonsense, missense, codon-deletion and canonical splice acceptor/donor variants. **d** SpliceAI Delta score distributions for canonical splice acceptor/donor variants, split by SGE functional class. Kruskal–Wallis $p = 0.002$. Compared to SGE-unchanged variants: FD: Fast-depleting $n = 130$, Dunn's BH-corrected FDR = 0.001, SD: Slow-depleting

$n = 35$, FDR = 0.3, U: SGE-unchanged $n = 51$ E: SGE-enriched $n = 8$ FDR = 0.66. Internal boxplots within each violinplot show median and interquartile range (IQR), whiskers denote 1.5xIQR. **e** SpliceAI Delta score distributions for intronic variants outside canonical splice sites split by SGE functional class. FD: Fast-depleting $n = 38$, SD: Slow-depleting $n = 66$, U: SGE-unchanged $n = 1677$, E: SGE-enriched $n = 104$. Internal boxplots within each violinplot show median and interquartile range (IQR), whiskers denote 1.5xIQR. **f** cLFC-trend for intronic and synonymous SNVs within 2 bp of the end of the exon plotted according to position relative to the splice site. Triangles denote pyrimidine to purine variants (Py > Pu). Source data are provided as a Source Data file.

canonical splice variants (Fig. 4c) compared to 19% (41/211) of splice variants in other exons ($X^2$ $p$ = 0.00002). Exon 3 is of phase 0 and lies outside of the functional *DDX3X* domains. Splice variants which induce exon-skipping of exon 3 may therefore produce an in-frame protein which retains helicase function (Fig. 4c). Whether this reflects a tolerated alternative splicing remains to be tested experimentally. Two such endogenous isoforms that lack exon 3 (ENST00000457138.6 and ENST00000631641.1) have been reported, but appear only weakly expressed in GTEx.

Outside canonical splice sites, 6% (105/1886) of intronic variants were SGE-depleted and 6% (104/1886) were SGE-enriched, with variants closer to exons being less likely to be SGE-unchanged (Fig. 4e, f). Both SGE-depleted and SGE-enriched intronic variants had higher SpliceAI scores compared to SGE-unchanged intronic variants, with fast-depleting variants having by far the highest SpliceAI scores. We characterised 87 variants in the 5′ UTR within 25 bp of the start codon and identified one SGE-depleted UTR variant predicted to generate an upstream, out-of-frame, start codon.

## Comparison to variants observed in clinical and population cohorts

We identified 536 *DDX3X* coding, canonical splice site and near-exon variants in UKBB or GnomAD, and 239 *DDX3X* variants in individuals with NDD from ClinVar, DECIPHER, the 100,000 Genomes Project and publications[9,26], including variants seen in males and females, with differing clinical interpretations and with differing inheritance.

Ninety-seven percent (521/536) of variants observed in UKBB and GnomAD were SGE-unchanged, two variants were slow-depleting and 13 were SGE-enriched. Variants seen in NDD cases were 72% (173/239) SGE-depleted, 3% (8/239) SGE-enriched and 24% (58/239) SGE-unchanged (Fig. 5a). All NDD variants clinically interpreted to be benign were SGE-unchanged, whereas only 13% (24/181) of those clinically interpreted to be pathogenic or likely pathogenic were SGE-

unchanged, with 85% (154/181) being SGE-depleted. We noted a strong sex-difference: all but one 'pathogenic' variant in males (8/9) were SGE-unchanged variants whereas only (10%, 12/117), of pathogenic variants in females were SGE-unchanged (Fig. 5a, b). We did not observe any correlation of phenotypic severity with the type or functional class of pathogenic DDX3X variant (Fig. 5b–f and Supplementary Results).

## Training and assessing a machine learning classifier for NDD-relevance

Supervised machine learning of SGE data, trained using likely pathogenic and benign variants, may increase accuracy to identify functionally abnormal *DDX3X* variants of NDD-relevance, compared to the unsupervised functional classification described above. We tested linear and Random Forest machine learning models. In all, 80% of likely pathogenic/pathogenic and likely benign/GnomAD/UKBB variants were used for training and the remainder used for testing. The Random Forest classifier (Figs. 6a, b and S6) only marginally outperformed linear models (Supplementary data S2), suggesting that model performance is driven by the data, rather than the model.

Random Forest model performance estimated from 'test variants', using a posterior probability threshold of 50%, was 97.1% sensitivity and 100% specificity (Table 1), due to the discordant classification of a single variant. Clinically interpreted variants may contain false positives, and therefore this performance may be a minimum estimate of sensitivity. SGE data outperformed in silico predictors of variant effects (Fig. 6d, e), which lack sensitivity at clinically-recommended thresholds[27] (Table 1) and lack specificity at commonly employed less stringent thresholds (Supplementary data S3). In clinical diagnostic workflows, evidence of variant effect from functional assays and in silico predictors is scored independently. Therefore, to enable diagnostic usage of our data, our classifier was intentionally derived only from the experimental SGE data and excluded in silico-derived

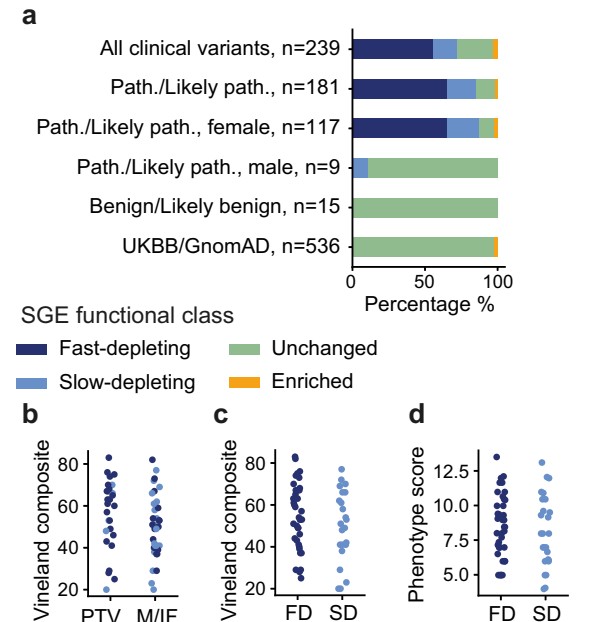

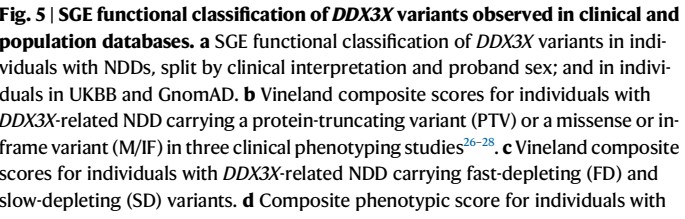

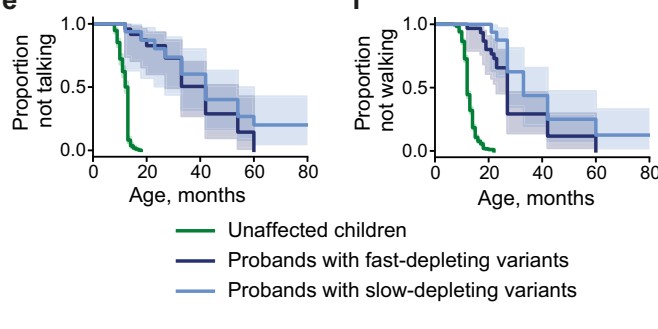

**Fig. 5 | SGE functional classification of *DDX3X* variants observed in clinical and population databases. a** SGE functional classification of *DDX3X* variants in individuals with NDDs, split by clinical interpretation and proband sex; and in individuals in UKBB and GnomAD. **b** Vineland composite scores for individuals with *DDX3X*-related NDD carrying a protein-truncating variant (PTV) or a missense or in-frame variant (M/IF) in three clinical phenotyping studies[26–28]. **c** Vineland composite scores for individuals with *DDX3X*-related NDD carrying fast-depleting (FD) and slow-depleting (SD) variants. **d** Composite phenotypic score for individuals with

*DDX3X*-related NDD carrying fast-depleting and slow-depleting variants. **e&f** Age at which first words (**e**) and first independent steps (**f**) were taken for individuals with *DDX3X*-related NDD carrying fast-depleting variants and slow-depleting variants, compared to children without an NDD. Number of individuals: First words: $n$ = 24 fast-depleting variants, $n$ = 16 slow-depleting variants; first steps: $n$ = 31 fast-depleting variants, $n$ = 17 slow-depleting variants. Source data are provided as a Source Data file.

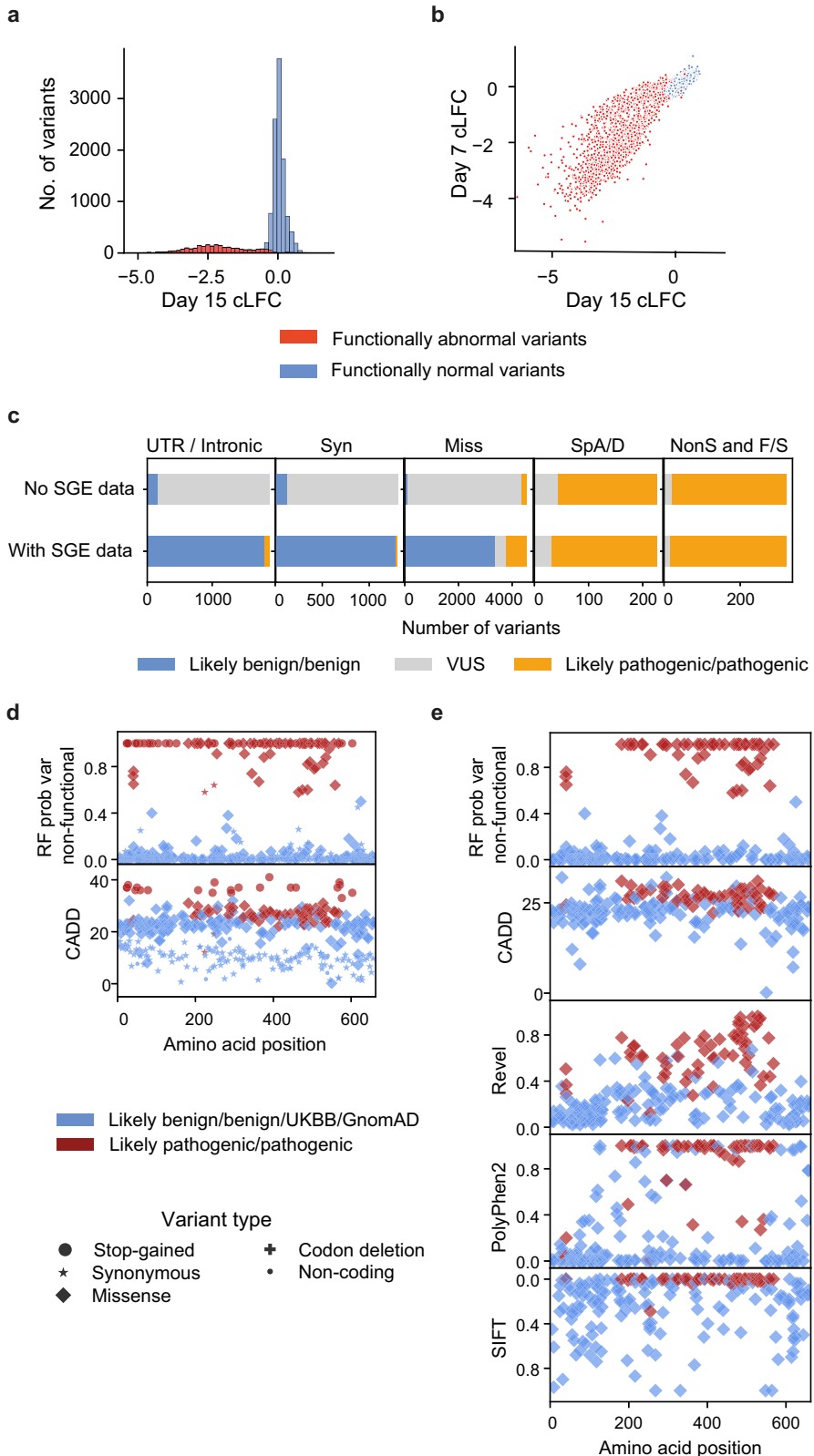

**Fig. 6 | Performance of a machine learning classifier of NDD-relevance. a** Day 15 cLFC of variant abundance. **b** Day 7 and Day 15 cLFC of variant abundance coloured by variants' NDD-relevance. **c** Modelling the impact of SGE data on *DDX3X* clinical variant interpretation for a hypothetical female patient with moderate intellectual disability for a variant of unknown inheritance status. NS nonsense, F/S frameshift, Splice A/D canonical splice acceptor/donor sites. **d**, **e** Comparison of in silico variant effect predictor scores and Random Forest classifier posterior probability for (**d**) all likely pathogenic/pathogenic, likely benign and GnomAD/UKBB variants and (**e**) missense likely pathogenic/pathogenic, likely benign and GnomAD/UKBB variants. Source data are provided as a Source Data file.

**Table 1 | Comparison of SGE supervised classifier to commonly used in silico predictors of variant effects**

|  | ROC AUC | TP | TN | FP | FN | Sensitivity (%) | Specificity (%) | PPV (%) | NPV (%) |
|---|---|---|---|---|---|---|---|---|---|
| **All variants** |  |  |  |  |  |  |  |  |  |
| SGE (RF test vars only) | 0.986 | 33 | 108 | 0 | 1 | 97.1 | 100 | 100 | 99.1 |
| SGE (all truth vars) | 0.997 | 167 | 539 | 0 | 1 | 99.4 | 100 | 100 | 99.8 |
| CADD | 0.971 | 108 | 514 | 22 | 21 | 83.7 | 95.9 | 83.1 | 96.1 |
| **Missense variants only** |  |  |  |  |  |  |  |  |  |
| SGE (RF test vars only) | 0.970 | 16 | 29 | 0 | 1 | 94.1 | 100 | 100 | 96.7 |
| SGE (all truth vars) | 0.995 | 81 | 159 | 0 | 1 | 98.7 | 100 | 100 | 99.4 |
| CADD | 0.898 | 65 | 138 | 21 | 17 | 79.3 | 86.8 | 75.6 | 89.0 |
| REVEL | 0.959 | 21 | 159 | 0 | 61 | 25.6 | 100 | 100 | 72.3 |
| SIFT | 0.927 | 62 | 149 | 10 | 20 | 75.6 | 93.7 | 86.1 | 88.2 |
| Poly Phen2 | 0.916 | 55 | 150 | 9 | 27 | 67.1 | 94.3 | 85.9 | 84.7 |

*ROC AUC* area under the Receiver-operator characteristic (ROC) curve, *TP* the number of test-positive true-positive variants, *TN* the number of test-negative true-negative variants, *FP* the number of test-positive true-negative variants, *FN* the number of test-negative true-positive variants, *SGE (RF test vars only)* calculated from truth set variants not employed in the training of the classifier, *SGE (all truth vars)* calculated from all truth set variants, including variants employed for training of the classifier. In silico variant effect prediction algorithm thresholds were chosen according to the minimum thresholds required for utilisation of scores as "supporting" evidence of variant effect identified by Pejaver et al. [27].

information. However, when CADD scores were incorporated within the model this slightly reduced its performance (Supplementary data S2).

We modelled the utility that this SGE-based classifier of NDD-relevant functional abnormality might have within existing frameworks for diagnostic variant interpretation (Fig. 6c) for a hypothetical female patient with moderate intellectual disability, under two scenarios: where the variant is known to be de novo (Supplementary data S4), and where inheritance status is unknown (Fig. 6c and Supplementary data S5). A Random Forest model posterior probability threshold of 50% was used. Variant classification was performed for all possible synonymous, missense, nonsense and canonical splice acceptor/donor variants and for frameshift, intronic and UTR variants included in the SGE libraries. The number of missense VUS of unknown inheritance was reduced by 91%, from 4282 to 393 variants, with an over fourfold increase in the number of likely pathogenic/pathogenic variants (178–783) and an over 30-fold increase in the number of likely benign/benign variants (100–3384; Supplementary data S5). For synonymous and non-coding variants, incorporation of SGE data reduced the number of VUS by over 99%, with the majority of variants being reclassified as benign. However, 14 synonymous and 73 non-coding (intronic and UTR) variants were reclassified as likely pathogenic/pathogenic (Fig. 6c and Supplementary data S5). The inclusion of SGE information had much less effect on the classification of predicted loss-of-function variants (nonsense, frameshift and canonical splice-site variants), in part as functional data supporting a benign interpretation can be outweighed by the substantial weighting accorded to these variant annotations under current ACMG guidelines [27].

We applied this NDD-relevant functional classifier to 23 rare non-synonymous *DDX3X* variants of unknown inheritance status from the DDD study [28]; 39% (9/23) of these variants were classified as being functionally abnormal (8/9 in female probands), and 61% as functionally normal. Although our data show ~100% specificity, we accept that in some clinical decision-making contexts a posterior-probability of variant function higher than 50% may be desirable. Therefore, we propose that for clinical use a posterior probability of > 90% be considered high confidence, and posterior probability 50–90% be considered intermediate confidence (Supplementary data S6 and Fig S7). 88% (1488/1689) of all DDX3X variants predicted to be functionally abnormal are of high confidence (posterior probability > 90%). In all, 259/280 stop-gained variants are predicted to be functionally abnormal with high confidence. The remaining 21 are predicted in silico to escape NMD. 17% (788/4560) of missense variants are predicted to be functionally abnormal, 642/788 (81%) with high confidence. 73% (146/

201) of intermediate confidence variants are missense variants (Fig. S7).

It is possible to estimate the proportion of all missense variants that are likely to be pathogenic for a given haploinsufficient disease by comparing the relative excesses in large disease cohorts of missense and nonsense de novo mutations (compared to the numbers expected under a null germline mutational model). Using the data from over 31,000 NDD families [9], we estimated that ~17.5% (95% CI 10.9–29.6%) of missense variants in DDX3X are likely to be pathogenic for NDDs. This corresponds closely to the 17% of missense variants that we predict to be functionally abnormal using the NDD-relevant functional classifier.

### DDX3X in cancer

Putative driver variants in *DDX3X* have been reported in diverse cancer types including medulloblastoma[13–15], various lymphoid tumours[29–34], and melanoma[35]. However, *DDX3X* has been classified as both a tumour suppressor and an oncogene[13,16,17,36]. We analysed *DDX3X* non-synonymous variants identified in 90,279 cancers[37,38], stratified according to whether *DDX3X* has been identified as a putative driver gene[39] in the cancer-type of origin. The proportion of fast and slow-depleting variants in *DDX3X*-driver cancers is 2.5 and 2.4x greater than in *DDX3X*-non-driver cancers ($X^2$ $p = 9.0 \times 10^{-5}$, $9.6 \times 10^{-5}$, respectively), while the proportion of SGE-enriched variants observed in these cancers is not significantly different ($X^2$ $p = 0.41$; Fig. 7a). We also applied dNdScv, for the subset of cancers for which synonymous variant data are available[40], to estimate the proportion of *DDX3X* missense variants acting as drivers in different cancer types, and to test how well this correlates with the proportion of these variants that appear functionally abnormal in our SGE assay, while correcting for mutational differences between cancers. In 33 different cancer types[41], we found that the proportion of somatic missense mutations estimated to be driver variants was highly correlated with the proportion of those variants that are SGE-depleted, but not with the proportion that was SGE-enriched (Figs. 7b, c and S8). The somatic missense mutations that were SGE-depleted could almost entirely account for the number of expected driver mutations per cancer. Notably, in medulloblastoma 99.7% (95% CI 99.1–100) of *DDX3X* missense variants are estimated to be drivers, and all are SGE-depleted ($n = 16$). Together, these data support the argument that *DDX3X* predominantly acts as a tumour suppressor gene across multiple tumour types.

## Discussion

We have characterised the relative abundance of 12,776 exonic and near-exonic variants in *DDX3X* over a timecourse of cell culture, identifying three classes of functionally abnormal variants: rapidly-

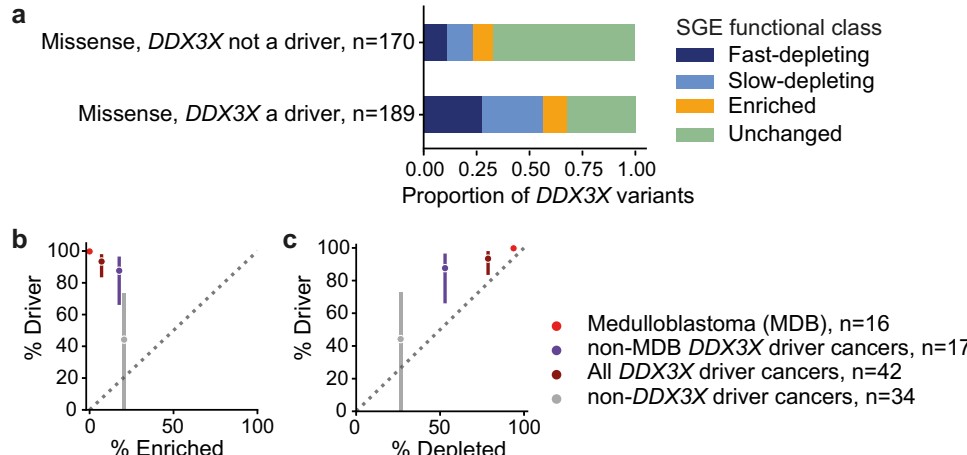

**Fig. 7 | SGE functional classification of *DDX3X* variants observed in cancers.**
**a** The proportion of SGE functional classes in *DDX3X* missense variants observed in cancers stratified by whether or not *DDX3X* had been identified as a putative driver gene in each cancer type. **b** comparing the proportion of missense variants classified as SGE-enriched with the estimated percentage of missense variants that are drivers, in different sets of cancer types. **c** comparing the proportion of missense variants classified as SGE-depleted with the estimated percentage of missense variants that are drivers, in different sets of cancer types. Error bars in **b**, **c** show 95% confidence intervals. Source data are provided as a Source Data file.

depleting, slowly-depleting and enriched. This map of variant effects allows us to identify functionally abnormal variants that might be expected to be functionally normal (e.g. synonymous, intronic and UTR variants) and functionally normal variants expected to be functionally abnormal (e.g. at canonical splice sites), in addition to identifying the minority (17%) of missense variants which are functionally abnormal. The systematic inclusion of codon deletions demonstrated that this class of variants is, on average, more damaging than missense variants, and should aid the clinical interpretation of in-frame indels.

A supervised classifier to identify functionally abnormal variants of NDD-relevance had higher accuracy in discriminating between pathogenic and benign variants than a range of in silico tools in common clinical usage, and resolves the majority of potential VUS in *DDX3X* (Supplementary data S4 and S5). This classifier only identified a single de novo pathogenic/likely pathogenic variant in a female with NDD as being functionally normal. This deposited missense variant (ClinVar ID 975674) was interpreted under the 2015 ACMG guidelines[8,40], but would likely be interpreted as a VUS following the 2018 ClinGen revised variant interpretation guidelines[8,41]. Unlike in females, variants interpreted to be pathogenic or likely pathogenic in males were almost all functionally normal in this assay. This could be because (i) the 'pathogenic' variants in males are not actually disease-relevant, (ii) the 'pathogenic' variants in males are disease-relevant and operate via a mechanism examined in this assay, but the assay is insufficiently sensitive, or (iii) the 'pathogenic' variants in males are disease-relevant but operate via a different mechanism not examined in this assay. Martin et al. previously reported that the statistical evidence for males with NDDs having an excess of non-synonymous rare variants in *DDX3X* compared to controls did not reach genome-wide significance[12]. The concordance between prior statistical genetic analyses and the functional data presented here suggests that further evidence is needed to support the association between damaging variants in *DDX3X* and NDDs in males.

SGE-depleted variants (but not SGE-enriched variants) were significantly over-represented in cancer types in which *DDX3X* has been shown to be a driver gene, and could account for almost all of the excess non-synonymous *DDX3X* somatic mutations seen in these cancers. This suggests that *DDX3X* predominantly operates as a tumour suppressor gene, and that this variant effect map identifies almost all of the cancer driver mutations. SGE-depleted variants are also over-

represented in cancer types in which *DDX3X* has not yet been conclusively demonstrated to be a driver gene, suggesting that *DDX3X* may play a tumour suppressor role in a wider range of cancers than currently appreciated, and that integrating this variant effect map should increase power in analyses of somatic mutation enrichment.

Further investigation is needed into the functionally abnormal enriched variants, to establish the mechanism of action of these variants, what their phenotypic consequences might be, and how this results in purifying selection acting on them.

We identified different kinetics of depletion among functionally abnormal *DDX3X* variants. Variants predicted to be most damaging to protein function such as nonsense, frameshift and splice-site variants tend to be fast-depleting, while missense and non-coding variants are over-represented in the slow-depleting class (Fig. 4c). Residues which when deleted or mutated result in fast-depleting SGE kinetics are found deeper within the DDX3X protein (Figs. 3a and S2e, and are enriched at interaction interfaces (Fig. 3b–d and Supplementary movies 1 and 2). However, we did not detect any phenotypic differences between patients carrying fast and slow-depleting variants, indicating that cell culture kinetics are not a reliable predictor of neurodevelopmental outcomes. We also did not observe different adaptive ability between patients carrying missense versus PTV variants, in contrast to what was reported previously[26,42]. It may be that phenotypic variability in patients with *DDX3X*-related neurodevelopmental disorder is dominated by other factors, for example, variable escape from[43-45] and skewing of X-inactivation[8,46].

Two previous SGE-based variant effect maps (for *BRCA1* and *CARD11*[4,47]) did not include all coding exons and characterised fewer variants than here for *DDX3X*. Nevertheless, all three SGE studies[4,47] have shown that a MAVE is more specific than in silico metrics in discriminating between pathogenic and benign variants. This demonstrates the limitations of evolutionary constraint-based metrics which cannot distinguish between different mechanisms of variant effect to identify disease-relevant variant subtypes. This may explain why in silico metrics of variant effect often have high sensitivity but limited specificity, (Fig. 6d, e, Table 1 and Supplementary data S3)[27]. MAVEs are therefore likely to be of particular value when applied to genes where purifying selection results from different mechanisms.

HAP1 cells are derived from a chronic myelogenous leukaemia line and lack apparent pathophysiological relevance to NDDs. This

suggests that, for some genes at least, the functional impact of variation can be protein-intrinsic and relatively agnostic to cell-type. With readily accessible databases of pathogenic and benign variation, it is possible to validate the utility of variant effect maps empirically and not rely on a priori biological intuition or expert consensus[48] with regard to the relevance of a given model system.

Our screen was performed with two sgRNAs per exon and with 5 timepoints. We have reanalysed the data by dropping an sgRNA and multiple timepoints in the dataset to test whether we could simplify the experiment without affecting the sensitivity. We observe a ROC-AUC of 0.988 when using only 3 timepoints, and 0.97–0.99 when using only 1 sgRNA per exon. This suggests that SGE could potentially be scaled down to a single sgRNA per target with 3 timepoints without compromising data quality or clinical utility.

There are 398 genes associated with developmental disorders or cancer (DDG2P - www.ebi.ac.uk/gene2phenotype; Cancer Gene Census - https://cancer.sanger.ac.uk/census) that appear to be essential for HAP1 cell growth[18,19] and have a pathophysiological loss-of-function mechanism. New techniques will be required to assess genes with pathophysiological altered-function effects, and genes not essential in HAP1 cells. Generating variant effect maps for hundreds of genes will require improvements to the scalability of SGE, however, this will be a collective, international endeavour (www.varianteffect.org) and open sharing of resources, protocols, code and data will propel progress.

## Methods

### SGE HDR oligo design and variant annotation
The HDR-library oligos (Supplementary data S7 and S8) were designed with a custom script. In short, the coding sequence of each exon and its 25 bp upstream and 15 bp downstream were extracted from the hg38 reference genome. Transcript NM_001356.4 was used for the design of the codon-related variants. Synonymous mutations at the sgRNA protospacer and PAM were then introduced manually to the extracted sequences. We designed two HDR plasmid libraries for each exon with the same set of variants of interest but with a different set of synonymous PAM/protospacer mutations. Then, the following oligonucleotides were designed: every SNV at every base position; tiled two base-pair deletions in the UTRs and intronic regions; in-frame deletion of every possible codon; and at least one redundant codon for each SNV at each base position, if possible. All indels shorter than 50 bp observed in these regions in GnomAD, ClinVar or DECIPHER were added manually to the final library pool. Illumina P5/P7 sequence and the homology sequences for Gibson cloning were appended at both ends of the oligos. All designed HDR-library oligos had at most 300 bases and were synthesised by TWIST Bioscience. HDR-library-associated variant call format (VCF) files were generated by re-running the same design with VaLiAnT[49]. VaLiAnT VCFs were then used to generate annotation files through VEP[50].

### TWIST oligo library preparation
The step-by-step protocols used in this study were deposited to Github (https://github.com/HurlesGroupSanger/Saturation_Genome_Editing/tree/main/Wetlab_protocols). The TWIST HDR-library oligo pools, consisting of designed HDR-library oligos from 17 exons, were dissolved in water at 10 ng/uL. 100 ng of the HDR-library oligo pools were amplified by P5 and P7 primers for 18 cycles with an annealing temperature of 60 °C in $2 \times 100\,\mu L$ reactions. The designed variant HDR libraries of each exon were then extracted from the amplified pools by PCR with exon-specific primers. The primer sequences for each exon of *DDX3X* are listed in Supplementary data S9. All primer oligos used in this study were purchased from IDT. 2X KAPA HiFi Hotstart ReadyMix (KAPA Biosystems) was used for all PCR reactions in this study, according to the manufacturer's protocol.

### sgRNA and HDR plasmid preparation
pMin-U6-ccdb-hPGK-puro (Supplementary data S10) was used for both sgRNA plasmid and HDR template library plasmid constructs. For sgRNA cloning, the annealed sgRNA oligo was cloned into the BbsI-HF (NEB) digested pMin plasmid. For HDR template library plasmid cloning, a genomic sequence of 800 bp upstream and downstream of a *DDX3X* exon was first amplified from HAP1 gDNA and followed by cloning with Gibson assembly (NEB) into the NotI-HF (NEB) and SbfI-HF (NEB) digested pMin plasmid. The homology-arm plasmid sequences were validated by Sanger sequencing (Eurofins). The validated plasmid was linearised by PCR and was assembled with the TWIST HDR template library by NEBuilder® HiFi DNA Assembly kit (NEB). The reaction was then transformed into Stellar chemically competent cells (Takara) according to the manufacturer's protocol. The HDR plasmid libraries were subjected to maxiprep (Qiagen) and quality control by Illumina Miseq with $2 \times 280$ pair-end sequencing. The sgRNA oligo sequences are listed in Supplementary data S11. Primer sequences for preparation of the HDR template libraries are listed in Supplementary data S12.

### HAP1 cell bank preparation
The HAP1 *LIG4* knock-out (KO) clone (product-ID HZGHC000759c005) which carries a 10-base deletion at the *LIG4* exon 3 gene locus (hg38: Chr13:108,210,833-108,210,842) (Fig. S9a), was purchased from Horizon Discovery and cultured according to the manufacturer's protocol. This HAP1 line was then transduced with the pKLV2-EF1a-BsdCas9-W (Addgene, 67978) lentivirus and was selected in 10 µg/mL blasticidin (Gibco) to generate *LIG4*-KO Cas9-expressing HAP1 cells. The Cas9-positive cells were then stained with 10 µg/mL of Hoestch 33342 (ThermoFisher Scientific) followed by sorting of the haploid G1 population with an XDP FACS sorter. The sorted haploid cells were expanded in the presence of 10 µg/mL of blasticidin for one week and were then banked in liquid nitrogen. The cells were sent for mycoplasma tests and karyotyping (Fig. S9b).

### Ploidy assay
HAP1 cells were arrested at metaphase by treatment with 0.1 µM nocodazole (Biovision) for 14 h in a 37 °C incubator. The treated cells were then dissociated with TrypLE (Thermo Fisher) and fixed with 80% ethanol. The fixed cells were treated with 0.1% TritonX-100 in PBS. $5 \times 10^5$ cells were then resuspended in 500 µL of 0.1% TritonX-100/PBS with DAPI at 1 µg/mL and incubated at room temperature for 30 min before performing the FACS assay. Singlet cells were gated for the analysis. Haploid and diploid cell populations were separated using the DAPI channel (Fig. S10c).

### Saturation genome editing
HAP1 cells were thawed and expanded 1 week before each screen started. In all, $8 \times 10^5$ of HAP1 cells were seeded in each well of a 12-well plate one day before transfection (Day −1). In total, 6–12 wells were used for each biological replicate in the SGE. In all, 2 µg of the sgRNA plasmid and 4 µg of the HDR template library plasmid were transfected into each well with 3.6 µL of Xfect transfection reagent and 100 µL of Xfect buffer (Takara). Media change was performed after 4 h of incubation at 37 °C. The cells were selected with 3 µg/mL of puromycin (Gibco) for two days, from 24 to 72 h after transfection (Day 1 and Day 2). The cells were passaged on Day 3 and were seeded into two T75 flasks. One of the flasks was harvested on Day 4, and the other flask was passaged on Day 5. The cells were then passaged every two days by re-seeding 10-25 million cells in 1–3 T150 flasks until Day 21. Cell pellets were harvested on Days 7, 11, 15 and 21. The cells were maintained with at least 10,000X library coverage for every passage. The genomic DNA (gDNA) was harvested by Qiagen DNeasy Blood and Tissue kit or Qiagen AllPrep DNA/RNA mini kit. SGE-targeted genomic loci were amplified by PCR and were sent for sequencing on Illumina Hiseq 2500 Rapid platform with $1 \times 300$ single-end.

## Sequencing library preparation

HAP1 genomic DNA (gDNA) was harvested by Qiagen DNeasy Blood and Tissue kit or Qiagen AllPrep DNA/RNA mini kit. 3-step nested PCR was performed to amplify the targeted exon from the HAP1 gDNA. To avoid amplifying the HDR template library plasmid that may persist transiently in the cell culture, the first-step PCR primers were designed outside the homology arm sequences of the targeted exons. For the second-step PCR, Illumina adaptor sequences were appended at the 5′ end of the target-specific primer sequences. Illumina P5/P7 and index sequences were added during the third-step PCR. In all, 4 µg of gDNA was used for the first-step PCR (4 x 50 µLreaction). 100 ng of the first-step PCR product was used for the second-step PCR (4 x 50 µLreaction), and 25 ng of the second-step PCR product was used for the third-step PCR (1 x 50 µLreaction, 7 cycles). The PCR reactions were purified with QIAquick 96 PCR Purification Kit (Qiagen) or with AMPure XP bead (Beckman Coulter) and the PCR products were quantified either by nanodrop or by Qubit dsDNA HS assay kit (Thermo Fisher). The pooled library was quantified with KAPA Library Quantification Kit Illumina® Platforms (KAPA Biosystems) and was sequenced on an Illumina Hiseq 2500 Rapid platform with 1 ×300 single-end. The primers used for each exon's sequencing library preparation and its PCR cycling conditions are listed in Supplementary data S13.

## Calculation of variant abundance, log fold change and LFC-trend

Illumina adaptor sequences in the raw fastq file were removed by Trim-galore[51]. The exon sequences were then extracted with Tagdust2[52] by using the primer sequences as the adaptor sequences. Each of the unique sequences was counted and the count tables were processed by R. For each sgRNA/HDR template library, the count table of each replicate and timepoint were joined. The joined tables were used for the DESeq2 analysis[53]. Sequences with a total read count less than or equal to 10 were removed from the analysis. The default DESeq2 scaling factor was replaced by each sample's total count-normalisation factor. This removes the assumption that most of the tested events have no significant change, which is not necessarily true for an SGE experiment. Day4 was used as a reference baseline for the LFC calculation of each time point. For the LFC trend calculation, the same DESeq2 code was applied, except that the timepoints were changed to "numeric", such that the baseline Day 4 was set to "0" and Day 7, 11, 15, 21 were set as "3", "7", "11", "17" respectively. The sequences that matched designed oligos were retained for the subsequent analysis. To use the synonymous and intronic variants as an internal control, the median of the raw LFC/LFC-trend from the sequences annotated by VEP as "synonymous_variant" and "intronic_variant" were subtracted from the LFC/LFC-trend of each variant. To combine the results (calculate for cLFC/cLFC-trend) from two independent sgRNA screens within the same exon, Inverse-Variance-Weighted Average and Weighted Sum of Z-Scores were applied to the LFC/LFC-trend where the sequences referred to the same DNA variant[54].

## Combining the sgRNA1 and sgRNA2 data at the PAM-codon

The synonymous PAM/protospacer mutations could complicate the interpretation of the variant effect observed at that codon. Subsequent introduction of a SNV within the PAM-codon produces a MNV and may lead to misinterpretation of the results. For example, we may expect a nonsense mutation from the SNV that changes the wild-type codon, CGA (Arg), to TGA (Stop). However, if we have introduced a synonymous mutation prior to the SNV, we produce a missense mutation instead, CGG (Arg) to TGG (Trp). We discarded the PAM-codon-specific variants to avoid misinterpretation during the analysis. Data from the other sgRNA will replace the discarded variants at the PAM-codon. In this SGE study, the sgRNA1 library consists of 12,777 unique sequences; 309 are sgRNA1-library-specific. 155 of these variants were located at the PAM codon and were not used in the analysis. 154 of these variants can be rescued by the sgRNA2 library. 1 variant was an undesired

missense as described above. sgRNA2 library consists of 12,776 unique sequences, 308 are sgRNA2-library-specific. The sgRNA1 library can rescue all 154 variants within the PAM-codon. cLFC and cLFC-trend values for all variants tested are reported in Supplementary data S14.

## Protein structural analysis of DDX3X

PyMOL was used to visualise the Alpha-Fold2 predicted structure of DDX3X: AF-O00571-F1-model_v4AFO0057113[55]. The experimentally derived PDB structure 2db3 was used to superimpose the position of RNA, ATP and $Mg^{2+}$ by RMSD minimisation of the backbones in pyMOL. To visualise the distribution of variant functional classes the modal missense outcome was calculated for each residue.

Estimation of distance to protein centroid for each residue: for each residue the average position of the coordinates in each side chain was used as the residue centroid point. To estimate the DDX3X protein centroid the mean of all side chain centroids was calculated to approximate the centre of mass. To estimate the distance of each residue from the protein centroid the direct through-space distance between the side chain centroids and the protein centroid was calculated in ångströms.

Determining whether amino acid residues are buried versus exposed: the total solvent accessible surface area (units: ångström^2, Å2) (SASA), and the surface area of each residue was obtained from the POPScomp server (http://popscomp.org:3838/) for the following PDB structures: 5e7i, 5e7j, 2jgn, 6o5f, 4px9, 2i4i, 6cz5, 5e7m, 4pxa, and the Alpha-Fold2 predicted structure of DDX3X: AF-O00571-F1-model_v4AFO0057113[55]. SASA values were only considered between amino acid residues 133 and 585, as data were available from at least two structures across this region. The mean SASA and mean residue surface area across these structures were calculated. Then the proportion of accessible surface area was calculated (mean SASA/ mean residue surface area). A residue was considered to be "buried" if the proportion of accessible surface area was less than 25%. The proportion of accessible surface area values for DDX3X amino acids are given in Supplementary data S14.

Gibbs free energy of folding (ΔΔG) values for DDX3X were obtained from Mutfunc[56]. ΔΔG was calculated using FoldX from the crystal structure of human DDX3X (PDB 5e7i). DDX3X ΔΔG values are provided in Supplementary data S14.

## Amino acid conservation scores

Conservation scores for each amino acid in DDX3X were generated as follows: BLASTP[57] was run on the protein sequence of human DDX3X (UniProt ID O00571) against the UniRef90 database with the following parameters: --matrix BLOSUM62 --exp 10 --dropoff 0 --alignments 1000 --scores 1000 --gapopen 10 --gapext 1 --align 0 --filter F --async. In order to limit the amount of processing, 200 pairwise alignments with a minimum identity of 25% were selected (Supplementary data S15). A pile-up alignment was built, inserting gaps where necessary, using O00571 as the reference sequence. The Scorecons algorithm[20] (https://www.ebi.ac.uk/thornton-srv/databases/cgi-bin/valdar/scorecons_server.pl), was used to compute the residue conservation scores with the following modification: where a matched sequence is shorter than the search sequence the "gaps" at the N- and/or C-terminal ends were excluded. Scorecons conservation scores, and the amino acid variation at each residue position, are given in Supplementary data S16.

## Clinical variant curation

Clinical variants identified in the context of *DDX3X*-related neurodevelopmental syndrome were identified from the following sources: ClinVar (4th December 2020); DECIPHER (4th December 2020); Genomics England 100,000 genomes study (21st January 2021); and the literature[9,26]. Benign and likely benign variants were grouped together. To generate a high-confidence set of likely pathogenic de novo variants, the following variants were included; ClinVar: de novo

variants annotated as likely pathogenic or pathogenic, for which there were no conflicting interpretations; DECIPHER: de novo variants annotated as likely pathogenic or pathogenic; Genomics England 100,000 genomes study: de novo variants considered clinically relevant (Tier 1 and 2 variants); de novo variants identified in a recent meta-analysis of 31,000 developmental disorder exomes considered clinically relevant, and where these variants had not been clinically interpreted as of uncertain significance or benign[9]; de novo likely pathogenic or pathogenic variants reported in Lennox et al. [26]. As the information on proband's sex is not available for all of these variants, no filtering based on proband sex was performed. If a variant was curated into the list of de novo likely pathogenic/pathogenic variants, it was included, even if it was observed as an inherited variant or a VUS in another source. Variants were removed from the list of VUS if they had been reported as likely pathogenic/pathogenic or likely benign/benign by another source. See Supplementary data S17 for all clinical variants, together with information on their source and clinical interpretation. Where the same variant is observed in multiple repositories it is not possible to ascertain whether the variant is observed in the same or different probands.

## GnomAD and UKBB variant curation and PheWAS analysis

GnomAD *DDX3X* variants were downloaded on 21st January 2021. Variants from GnomAD v2.1.1 were lifted over to build hg38 using python liftover (https://github.com/jeremymcrae/liftover).

UKBB: data from all 454,787 individuals with available whole-exome sequencing from the UKBB was used for PheWAS analysis. For all other analyses, the previous 200,629 individual whole-exome dataset was used[58].

To identify variants in *DDX3X* observed among the UKBB[59] individuals, we queried GRCh38-aligned population-level VCF provided via the UKBB research access platform (showcase field 23148). Prior to analysis, we split multiallelic variants and left-corrected and normalised indel variants using bcftools[60]. We next applied genotype-level filtering where individual genotypes were set to null (i.e. "./.") if the genotype had a depth <7, genotype quality <20, was called as heterozygous in a genetically male individual, or a binomial test $p$-value for alternate versus reference reads for only heterozygous genotypes ≤ 0.001. If more than 50% of genotypes were missing for a given variant, that variant was filtered. All variants were then annotated with VEP v102 and assigned to a gene-based on the primary MANE select v0.97 transcript[61] with the most severe consequence. Following variant quality control and processing, we then selected 421,064 individuals of primarily European genetic ancestry for further analysis. We next queried quality-controlled variant call files for all variants within *DDX3X* (MANE transcript NM_001356.5) identified as SGE-enriched/SGE-depleted. This query yielded a total of 18 variants across 37 individuals, of which 16 and 2 were SGE-enriched or slow-depleting, respectively. Due to the low overall number of slow-depleting *DDX3X* variants, further analyses were limited to SGE-enriched variants.

To determine the phenotypic consequences of carrying such variants, we queried UKBB-provided complete health outcomes data (combines general practitioner records, hospital episode statistics, death records, and self-reported conditions), cancer registry data and a subset of binary and continuous phenotypes deemed likely relevant to *DDX3X* loss or gain-of-function (Supplementary data S1). We then tested whether carrying an 'enriched' *DDX3X* variant led to a significant increase in risk for any of the conditions or phenotypes outlined above. Models were implemented in pythonv3.7 using the 'statsmodels' package[62] with family set to 'binomial' or 'gaussian' if the trait was binary or continuous, respectively. All models were corrected for age, $age^2$, sex, WES sequencing batch, and the first ten genetic principal components as defined in Bycroft et al. [63] We corrected for 9 tests ($p < 5.6 \times 10^{-3}$) and found no significant associations.

## Analysis of observed vs expected numbers of variants in UKBB and GnomAD

A triplet-based neutral mutational model[23] was used to estimate the likelihood of each *DDX3X* SNV arising per generation. Analysis was limited to SNVs occurring within 10 bp of an exon-intron boundary to ensure exome data coverage. The sum of mutational probabilities for SNVs in each SGE functional class (SGE-unchanged, fast-depleting, slow-depleting, SGE-enriched) was calculated. Across UKBB and GnomAD there are 520 observations of *DDX3X* SNVs (counting a variant observed in both databases as two observations). To estimate the number of SNVs in each SGE functional class that would be expected if the SNVs arose according to the neutral mutational model, the total number of observations of *DDX3X* SNVs was multiplied by the summed mutational probability for each SGE functional class. Supplementary data S18 gives the mutational probability for each SNV in *DDX3X* as per the triplet-based neutral mutational model.

## Genotype-phenotype correlation

Vineland Adaptive Behaviour Composite scores from three studies were identified[26,42,64], Supplementary data S19. Where necessary, genomic coordinates were obtained for published patient variants using the Mutalyzer position converter tool (https://mutalyzer.nl/position-converter). If the variant was reported as a cDNA nucleotide change without specifying the ID of the relevant transcript[26], transcript NM_001356.4 was used to map the variant to genomic coordinates. Genomic coordinates were checked against reported amino acid changes. The Kolmogrov-Smirnov test (KS-test) indicated that Vineland composite scores were normally distributed, therefore two-tailed T-tests were performed. As these studies used a mixture of the Vineland 2nd and 3rd editions, ANCOVA analysis incorporating patient age and Vineland edition as covariates was performed. No association between the Vineland Adaptive Behaviour Composite score and whether a *DDX3X* variant was SGE-fast or SGE-slow depleting was identified. The following python packages were used for statistical analysis: two-tailed T-test and KS-tests were performed in SciPy[65]; ANCOVA was performed using pingouin[66]. In order to investigate broader phenotypes, a composite score was devised for the Lennox et al. cohort, encompassing brain MRI findings, neurological phenotype, cardiac findings, precocious puberty, the experience of seizures and behavioural assessment according to the scoring metric in Supplementary data S20. Where data was partially missing (for example, if a patient had not undergone an MRI), the cohort's median value for that aspect of the score was assigned to that individual. All phenotypic scores for the Lennox et al. cohort are available in Supplementary data S21. Milestone data for unaffected children were drawn from the following sources: Age of walking[67]; Age of talking: ages of healthy English-speaking children with up to 5 words extracted from http://wordbank.stanford.edu/[68] (a 'word' was considered to be any sound used with meaning, such as 'baa' for sheep). Supplementary data S22 and Supplementary data S23 give the age of walking and talking for DDD probands and unaffected children, respectively.

## Training and assessing a machine learning classifier for NDD-relevance

The following supervised classifiers, built using scikit-learn in Python[69], were trained using cLFC on day 7, 11 and 15: logistic regression, Linear Discriminant Analysis (LDA), Gaussian Naive Bayes and Random Forest. Variants observed in the GnomAD and UKBB population databases and benign/likely benign variants deposited in ClinVar and DECIPHER were used as 'negative truth set' variants ($n = 540$). 'positive truth set' variants ($n = 168$) were de novo pathogenic/likely pathogenic variants curated as described above. Stratified sampling was used to select 80% of these truth set variants to train the model. Model parameters were optimised as follows: Random Forest: default parameters were used

for bootstrap sample size; the number of features randomly sampled for each split point; the number of trees and tree depth, as modification of the default did not improve model performance. The class with the highest mean probability estimate across the trees in the forest model was taken as the model-predicted class. LDA: SVD, LSQR and Eigen solvers were tested with the best performance provided by SVD. Shrinkage was not used.

For each model, variants categorised with the positive truth set training variants will be referred to as functionally abnormal, and variants categorised with the negative truth set training variants will be referred to as functionally normal[70]. Model performance (Table 1, Supplementary data S2 and Supplementary data S24) was evaluated on the 20% of truth set variants that were not employed for model training. Sensitivity, specificity, positive and negative predictive values for each model were estimated. For the purposes of these calculations, true positive (TP) variants were variants predicted as non-functional by the clinical classifier, and present in the test set of de novo pathogenic/likely pathogenic variants. True negative (TN) variants were variants predicted as functional, and present in the test set of GnomAD/UKBB/clinical benign variants. GnomAD/UKBB/clinical benign variants called as non-functional were considered false positives (FP), while de novo likely/pathogenic variants called as functional were considered false negatives (FN).

Once Random Forest had been identified as the highest performing classification model, a variety of different data inputs were evaluated. Model sensitivity and specificity increased by incorporating log fold-change data from multiple timepoints. The best performing model incorporated variant LFC on Day 7, 11 and 15. The addition of variant log fold-change on Day 21 did not further improve the performance of the model (Supplementary data S24). Supplementary data S24 gives the average number of true and false positives and negatives, sensitivity, specificity, positive and negative predictive values of ten independent versions of each model, evaluated on the truth set test data.

## Comparison of SGE clinical classifier to in silico variant effect prediction programmes

Thresholds were chosen according to the minimum thresholds required for utilisation of in silico variant effect prediction scores as "supporting" evidence of variant effect identified by Pejaver et al. [27] as follows: SIFT < 0.0001: test-positive, SIFT ≥ 0.001: test-negative[71]. PolyPhen2 > 0.978: test-positive, PolyPhen2 ≤ 0.978: test-negative. REVEL > 0.773: test-positive, REVEL ≤ 0.773: test-negative, CADD > 25.3: test-positive, CADD ≤ 25.3: test-negative (Table 1). Default developer-recommended thresholds were used for in silico variant effect prediction tools as follows: SIFT < 0.05: test-positive, SIFT ≥ 0.05: test-negative. PolyPhen2 > 0.902: test-positive, PolyPhen2 ≤ 0.902: test-negative. REVEL > 0.5: test-positive, REVEL ≤ 0.5: test-negative, CADD > 20: test-positive, CADD ≤ 20: test-negative. Results are shown in Supplementary data S3.

## Estimation of the proportion of *DDX3X* missense variants pathogenic for DDX3X-related NDD

If all missense variants are pathogenic, then the proportion of missense:nonsense variants observed in a large disease cohort is expected to be equal to that predicted by the germline mutational model. According to the neutral mutational model, we expect 20.5x more missense than nonsense variants. Using the data from over 31,000 NDD families[9], 72 de novo missense and 20 de novo nonsense variants were observed. Therefore, we estimate that ~17.5% of missense variants in *DDX3X* are likely to be pathogenic for NDDs.

## Modelling the impact of SGE data on clinical variant interpretation

The Clinical Genome Resource (ClinGen) Sequence Variant Interpretation (SVI) Working Group has published guidelines for the incorporation of functional assays into clinical variant interpretation[48]. We evaluated the performance of our SGE data according to these guidelines and established that a variant with an SGE clinical classification of non-functional would merit application of the PS3 code during clinical variant interpretation (strong evidence that a variant is pathogenic), while a variant with an SGE clinical-classification of functional would merit application of the BS3 code during clinical variant interpretation (strong evidence that a variant is benign), see Supplementary data S25, adapted from Supplementary Table 1 of Brnich et al. [48] To assess the impact of including SGE data in clinical variant interpretation we performed in silico interpretation of all possible synonymous, missense, splice A/D and nonsense variants, and those non-coding and frameshift variants included in the SGE HDR template library in the context of each variant arising in a female proband with moderate intellectual disability. The exercise was performed twice, considering each variant as having arisen de novo (Supplementary data S4) or where the inheritance of that variant is unknown (Supplementary data S5). Codes were assigned as follows, according to the ClinGen SVI Guidelines:

BS1 criterion:

1. Moderate if variant not observed in either UKBB or GnomAD datasets.
2. Strong if allele frequency > $3.25 \times 10^{-7}$ in any of the major population groups of GnomAD or UKBB. The threshold was estimated as $3.25 \times 10^{-7}$ using http://cardiodb.org/allelefrequencyapp/[72] as follows:
   - Prevalence of DDX3X-related DD: 1/27700, calculated as 1% (population prevalence of ID) x prevalence of *DDX3X* de novo in the Kaplanis et al. (112/31058)[9], which is consistent with previous estimates.
   - Allelic heterogeneity: estimated as 0.09. In Kaplanis et al., the most common variants were observed 5 times, with a total of 112 different de novo *DDX3X* variants, 0.09 is the upper bound of the CI[9].
   - Genetic heterogeneity: 1 (only *DDX3X* has been reported to cause *DDX3X*-related neurodevelopmental syndrome).
   - Penetrance: 0.5 (likely to be an under-estimate).

PP3/BP4 criteria:

1. PP3 - supporting if SpliceAI Delta score > 0.7 or REVEL score > 0.7, or if REVEL score not available if CADD PHRED score > 25.
2. BP4 - supporting if SpliceAI Delta score <0.2 and if REVEL score <0.2, or if REVEL score not available, if CADD PHRED < 10.

PS2 criterion:

1. Supporting if the variant is not de novo but this variant has previously been reported as de novo and pathogenic or likely pathogenic.
2. Moderate if the variant is de novo and this variant has previously been reported as de novo and pathogenic or likely pathogenic.

PVS1 criterion

Nonsense and frameshift variants

1. Very strong if the variant is predicted to undergo NMD.
2. Strong if the variant is not predicted to undergo NMD, occurs distal to the DDX3X helicase domains, and results in truncation of > 10% of the protein. (*DDX3X* LOFs are not frequent in the general population, including in the last 10% of the protein).
3. Moderate if the variant is not predicted to undergo NMD, occurs distal to the DDX3X helicase domains, and results in truncation of <10% of the protein.

Canonical splice acceptor/donor variants (PP3 was not applied to avoid double-counting of information):

1. Very strong if the exon is not of phase 0 so exon skipping would be predicted to result in a frame-shift, and the variant is predicted to undergo NMD.
2. Strong if the exon is not of phase 0, the variant is not predicted to undergo NMD, and the resultant frame-shift caused by exon skipping would result in truncation of > 10% of the protein.
3. Moderate if the exon is not of phase 0, the variant is not predicted to undergo NMD, and the resultant frame-shift caused by exon skipping would result in truncation of <10% of the protein.
4. Moderate if the exon is of phase 0, the variant occurs outside of the regions encoding the DDX3X helicase domains. (All DDX3X exons are <10% of the length of the coding length).

The following variants were predicted to escape NMD:
- Occurring within the first 200 bp of the coding sequence of *DDX3X*.
- Occurring distal to 55 bp 5′ of the last coding splice junction (between exons 16 and 17) of *DDX3X*.

Missense variant-specific criteria:
1. PP2 - supporting for all variants, as missense variants are frequently pathogenic in DDX3X-related neurodevelopmental syndrome and DDX3X is known to be missense-constrained.
2. PS1 - strong, if the variant resulted in the same amino-acid change as a previously reported de novo pathogenic or likely pathogenic variant.
3. PM5 - moderate, if the variant resulted in a different amino acid at a residue with a previously reported de novo pathogenic or likely pathogenic variant.
4. PP5 - supporting if a non-de novo variant had been reported as pathogenic or likely pathogenic at this amino acid residue, regardless of the match of the amino acid change.

The quantitative scoring framework outlined by Tavtigian et al. [73] was used to sum across criteria and classify a variant as benign, likely benign, of uncertain significance, likely pathogenic or pathogenic.

### DDX3X variants in cancer

*DDX3X* missense variants in 90,279 independent cancer samples were obtained from cBioPortal[37,38] (Supplementary data S26) The IntOGen resource was used to identify whether *DDX3X* had been identified as a driver gene in this cancer type (https://www.intogen.org/search?gene=DDX3X)[39]. *DDX3X* was considered a driver gene if a minimum of two methods had identified it as such. *DDX3X* was considered to be not a driver if no methods had identified it as such. Thus, *DDX3X* was identified as a driver gene in medulloblastoma, pilocytic astrocytoma, chronic lymphoblastic lymphoma, and lymphoma. The proportion of fast and slow-depleting missense variants in *DDX3X*-driver and *DDX3X*-non-driver cancers were compared.

### dNdSCV analysis

dNdScv estimates the ratio of non-synonymous to synonymous variation (dN/dS ratio) to identify likely driver genes where there is an excess of non-synonymous variation. For a given cancer sample, dNdScv estimates the background mutation rate of each gene by incorporating the gene-specific synonymous mutation rate with the variation in mutation rates across genes and the epigenomic context, while controlling for genic sequence composition and mutational signatures. dNdScv can also be utilised to estimate the proportion of cancer-associated missense variants likely to be driver variants[40]. As *DDX3X* has been reported to act as a male-specific driver gene in some cancers[35,74], dNdScv was first run on male and female samples from each tumour type separately (Fig S8 and Supplementary data S27). Where non-synonymous substitutions in *DDX3X* occur at a significantly higher (corrected $p < 0.05$) than the expected rate, we considered

*DDX3X* to be a driver gene (Supplementary data S27). *DDX3X* was identified as a driver gene in medulloblastoma and lymphoid tumours in both sexes, melanoma of the skin in males only, and uterine corpus endometrial carcinoma and cervical squamous cell carcinoma/endocervical adenocarcinoma in females, consistent with the previous literature[13–15,29–32,34,35,74]. *DDX3X*-driver and *DDX3X*-non-driver datasets were pooled and dNdScv re-run to estimate the proportion of missense variants likely to be driver variants. It is likely that there are further tumours where *DDX3X* acts as a male-specific driver gene ($p$-value for *DDX3X* substitutions = 0.08, for indels = 0.001 in pooled male "*DDX3X*-non-driver" tumour samples) Supplementary data S28.

### Statistical analysis

The statistical test, $p$-value, sample size and the cutoff used for each analysis were stated in the text, figure legend or "Methods" section.

### Reporting summary

Further information on research design is available in the Nature Portfolio Reporting Summary linked to this article.

## Data availability

The raw DNA sequencing data of HAP1 cells isolated from the SGE experiment at days 4, 7, 11, 15 and 21 generated in this study were deposited to ENA with accession number PRJEB52929. The analysed data were deposited in MaveDB with accession number urn:mavedb:00000658, [https://www.mavedb.org/#/experiment-sets/urn:mavedb:00000658]. Both raw and analysed data are depositied in BioStudies with accession number S-BSST1013 [https://www.ebi.ac.uk/biostudies/studies/S-BSST1013?query=Saturation%20genome%20edi[…]es%20pathogenicity%20of%20germline%20and%20somatic%20variation]. The following open-access data sources were used in this manuscript: ClinVar [https://www.ncbi.nlm.nih.gov/clinvar]; DECIPHER [https://www.deciphergenomics.org/]; GnomAD [https://gnomad.broadinstitute.org/]; cBioportal [https://www.cbioportal.org/]; the TCGA Research Network: [https://www.cancer.gov/tcga]; IntOGen [https://www.intogen.org/search] and The GTEx project [https://gtexportal.org/home/]. The manuscript also uses data from the following sources, which are available to researchers on application: Genomics England [https://www.genomicsengland.co.uk/research/academic/join-gecip] and the UK Biobank [https://www.ukbiobank.ac.uk/]. Data used for the analyses described in this manuscript were obtained from the GTEx Portal on 02/02/23; cBioportal on 05/05/2022, UKBB on 20/04/2021 ClinVar on 04/12/2020, DECIPHER on 04/12/2020, Genomics England 100,000 genomes study on 21/01/2021, gnomAD v2.1.1 and gnomAD v3.1 on 21/01/2022. Source data are provided with this paper.

## Code availability

All code used in this study is available from the Github repository (https://github.com/HurlesGroupSanger/Saturation_Genome_Editing/tree/main/Codes).

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

## Acknowledgements

We thank the Cytometry Core Facility, Molecular Cytogenetic Team and High Throughput Sequencing Team at Wellcome Sanger Institute for the technical support in this study. We thank E. Delage for assistance with data deposition. We also thank L. Parts, N. Whiffin, G. Findlay, M. Gasperini, K. Samocha, J. Kaplanis, A. Bassett, J. Buxbaum, D. Grice, E. Sherr and members of the Hurles group and AVE alliance for useful discussion on data analysis. We thank F. Day for assistance in preparing cancer phenotype data derived from the UK Biobank. This research was made possible through access to the data and findings generated by the 100,000 Genomes Project. The 100,000 Genomes Project is managed by Genomics England Limited (a wholly owned company of the Department of Health and Social Care). The 100,000 Genomes Project is funded by the National Institute for Health Research and NHS England. Wellcome, Cancer Research UK and the Medical Research Council have also funded research infrastructure. The 100,000 Genomes Project uses data provided by patients and collected by the National Health Service as part of their care and support. The DDD study presents independent research commissioned by the Health Innovation Challenge Fund (grant no. HICF-1009-003). The full acknowledgements can be found online (www.ddduk.org/access.html). The Genotype-Tissue Expression (GTEx) Project was supported by the Common Fund of the Office of the Director of the National Institutes of Health, and by NCI, NHGRI, NHLBI, NIDA, NIMH, and NINDS. The data used for the analyses described in this manuscript were obtained from the GTEx Portal on 02/02/23. The results used in the analysis are in part based upon data generated by the TCGA Research Network: https://www.cancer.gov/tcga. PheWAS analyses were conducted using UKBiobank application 9905 (to J.R.B.P.). We thank the DDX3X Foundation and DDX3X Support UK for their engagement with and enthusiasm for this research. This study makes use of data generated by the DECIPHER community. A full list of centres who contributed to the generation of the data is available from https://deciphergenomics.org/about/stats and via email from contact@deciphergenomics.org. Funding for the DECIPHER project was provided by Wellcome. This work was supported by core Wellcome funding to the Wellcome Sanger Institute (grant reference number: 108413/A/15/D), and for the purpose of open access, the author has applied a CC BY public copyright licence to any Author Accepted Manuscript version arising from this submission. This work was also funded by a Clinical Lecturer Starter Grant from the Academy of Medical Sciences, the Wellcome Trust, the Medical Research Council, the British Heart Foundation, Versus Arthritis, Diabetes UK, the British Thoracic Society (Helen and Andrew Douglas bequest), and the Association of Physicians of Great Britain and Ireland: SGL023\1060 to E.J.R. E.J.G. and J.R.B.P. are funded by the Medical Research Council (Unit programs: MC_UU_12015/2, MC_UU_00006/2, MC_UU_12015/1, and MC_UU_00006/1).

## Author contributions

E.J.R., H.T., S.S.G. and M.E.H. developed the study concept. E.J.R., H.T., M.H.L.A., H.I., A.J.W., D.G. and S.S.G conducted the SGE experiments. E.J.R., H.T. and S.L. performed formal analyses. J.D. and E.J.R. performed the protein structure analysis. E.J.G. and J.R.B.P. performed the UKBB data analysis. E.J.R, F.A., I.M. and D.J.A. performed cancer-related data analysis and interpretation. E.J.R., A.K., E.N., H.V.F. and K.B. performed NDD-related data analysis and interpretation. E.J.R. and M.E.H. acquired funding to conduct this work. S.S.G and M.E.H. supervised all analyses and experiments. E.J.R. wrote an original draft, and E.J.R., H.T., S.S.G. and M.E.H. edited the manuscript with input from all authors.

## Competing interests

The authors declare no competing interests.
