## [Peer Review File · Nature Communications]

REVIEWER COMMENTS

Reviewer #1, expertise in genomics, bioinformatics, systems biology and neuro disease (Remarks to the Author):

In this work, saturation genome editing (SGE) was performed on all 17 coding exons of DDX3X. Analysis of SGE results over time-course data revealed four categories of variants – unchanged, enriched, fast-depleting, and slow-depleting – showing the complex landscape of variant effects. To gain more insights into each group of variants, variant categories from SGE were compared to the functional annotations (e.g., missense), deleterious scoring (e.g., CADD), protein structure (e.g., buried, protein domain), effect on splicing, population cohort (e.g., GnomAD), and clinical annotations (e.g., ClinVar). A Random Forest Classifier was trained using SGE data to demonstrate the utility of SGE in distinguishing pathogenic variants from benign ones in neurodevelopmental disorders. Finally, the authors analyzed SGE behavior of DDX3X variants in cancer to explore potential role of DDX3X as a tumor suppressor gene.

Overall, the study is exciting and can benefit from addressing the following comments:

1. Several parts of manuscript and figures need improvement in clarity:

a. Figure 2:

i. Please include color legend at the beginning of figure or for each panel.

ii. In panel c, what is the confidence interval? Is z-score calculated within each group or on all the data? Is the y-axis for each group directly comparable? The authors could consider normalizing each group to all start at 0 on Day 1.

iii. Panels f and g, is the comparison with enriched group not significant or omitted here?

b. Figure 5b, c: The 33 different cancer types are categorized into 4 groups (which seems to be overlapping groups?), do all cancer types in each group have the same % Enriched and % Depleted (x-axis)? How about instead showing a scatter plot with each point denoting each cancer type?

c. Random forest classifier: What exactly are input features? Is input vector of length 1 showing the cLFC value of the respective variant or of length 3 with LFC on Day 7, 11 and 15?

2. It's great seeing random forest classifier trained using SGE data outperform classification using other deleterious scores (e.g., CADD) at clinically recommended thresholds. However, it is possible that the recommended threshold does not maximize for performance in this specific task. Could authors please also show performance comparison when training random forest classifier using CADD, REVEL, SIFT, and PolyPhen2 scores? Also has linear classifiers been tried to demonstrate the necessity of using a more complex nonlinear model?

3. It is very interesting that two distinct modes of depletion (i.e., fast-depleting and slow-depleting) exist among variants. What are potential biological mechanisms causing this differentiation? Do fast and slow

depleting variants cluster at different parts on 3D structure from PDB or AI predictions (e.g., AlphaFold or RoseTTAFold)? Could this be caused by a distinction in protein-protein interactions affected by different groups of variants? Last but not the least, are there prognosis differences between fast-depleting and slow-depleting variants in cancer?

Reviewer #2, expertise in genome editing, functional genomics, variant functional characterisation and cancer genomics (Remarks to the Author):

DDX3X Review

Radford, Tan et al. present a herculean effort to generate functional scores for all possible SNVs and some deletions for DDX3X by saturation genome editing (SGE). This work is of great significance for multiple reasons 1) this is the largest and most comprehensive sequence - function map generated by SGE to date, 2) it solves most of the variant classification problem for an important gene underlying developmental delay in females, 3) the authors validate the results and translate them into evidence for immediate use in clinical variant interpretation workflows. The data are highly sensitive and specific for accurately predicting clinical outcomes. However, there are a few analyses, explanations and figure additions/modifications (listed below) that we believe may strengthen the results as presented and add another level of transparency for readers and consumers of this data. We believe this work should be accepted and published as soon as possible.

MAJOR COMMENTS

1. Using DESeq2 might make sense for this type of read count data, but is not a precedent for MAVE. DESeq2 is used for determining differentially expressed genes or chromatin peaks where one does not know the truth. Here, that is not the case, for an assay measuring LoF, most stop codons and most synonymous codons can be used as an internal truth set. Same for known pathogenic and benign variants. Both of these types of “truth” variants can be used to derive sensible thresholds for functional categories. Can you explain why DESeq2 is a good choice for this dataset?

A. As deployed there is no intermediate function category, why was that decision made?

B. What do the known pathogenic and benign variants look like when mapped onto Fig 2a or Fig 2b?

C. What do synonymous and nonsense variants look like when mapped onto Fig 2a or Fig 2b?

2. There are 6 measurements for each variant (2 guides X 3 technical replicates). What does the distribution error between those measurements look like? Do some target regions have higher error than others? What correlates with the error (e.g. read count? specific targets? gDNA quant?)? Can the

error be taken into account for the clinical validation (e.g. are you equally confident in each variant score or RF classification or should some be down weighted or tossed entirely)?

3. Another addition to fig 2 (or the supplement) would be a plot of LfC by MAF. Evolution has already done the experiment. It would be good to see the correlation by functional score and minor allele frequency from GnomAD/UKBB.

4. There is an enriched population of variants according to the rate of growth in Fig 2c. It seems as though these variants are enriched in exons 3, 6, 7. However, these exons are outside of the folded functional domains and don't seem to be important for protein function given the missense and deletion data. Can your team speculate on a proposed mechanism?

A. Like above, we think it would be beneficial to show the distribution of the LFC of the synonymous variants to show that the enriched variants are not technical noise.

5. Random forest model: it would be nice to map the random forest classes back onto either the scatter plot or histogram like fig 2a, b.

6. An HGVS string for the variant is missing from the final data set.

7. The discussion ends with a call to arms for applying MAVE or SGE to more genes to generate functional evidence that can be used to interpret variants. Can you speculate on what that would take given what your team learned during the course of these experiments?

Two guides per target were used here. Is that necessary?

Can these experiments be scaled down?

What level of accuracy is needed for use in the clinic?

Is LoF in HAP1 an appropriate model for all DDD genes?

MINOR COMMENTS

1. The variant effect map is buried in Fig. S2a. The map in the supplement is new for the MAVE community, but that's progress! Fig S2 isn't terribly clear, however.

A. Maybe make a separate map of the codon deletions (they are hard to see)?

B. Maybe make it its own figure instead of smooshing it in with Fig S2b?

C. We know we're biased, but we really like the sequence function map in Findlay et al (Figure 4). Since the sequence function map is in the supplement anyway, why not take up the space?

D. It would also be great to include the exon and functional motif cartoons in the main text figure so readers (reviewers) don't have to flip back and forth.

2. There are 3 different protocols for tissue culture on your github

3. We think it would be beneficial to map the variant effects onto the protein structure to augment the genetic data. We feel like a more robust discussion of the effects of variation on protein structure/function could be added.

4. More analysis of variant effect predictors. Are there predictors that do better or worse on different parts of the protein?

5. In page 9 paragraph 2, it is stated that in-frame exon 3 has a higher percentage of SGE-unchanged variants than other exons. Are you suggesting the protein is functional without exon 3? Has alternative splicing of DDX3X been identified in RNA-seq data?

6. The discussion of the "wrong" SGE results for DDX3X variants in males. We're not experts in DDD but it seems as though those are highly likely to be incorrect classifications. The self flagellation in the discussion is not warranted. (Any chance y'all can take a second look at the genomes/exomes for those patients?)

7. DDX3Y isn't expressed in HAP1 cells since HAP1 lacks the Y chromosome. Given DDX3X and DDX3Y share 92% homology, could you include some brief discussion on this model towards the study of DDX3X-related intellectual disability in males?

Reviewer #3, expertise in machine learning, computational genomics, variant effect prediction and cancer genomics (Remarks to the Author):

The manuscript by Radford, Hurles, and colleagues describes multiplexed assays of variant effect (MAVS) of DDX3X, one of the most important risk genes of neurodevelopmental disorders. The in vitro functional assessment of DDX3X variants was done in HAP1 cells, a system used previously in MAVS of genes like BRCA1. About 3000 functionally abnormal variants were identified based on abundance at multiple time points. These variants can be grouped in 3 categories that may represent loss of function,

hypo-morphic, and possibly gain of function. The manuscript has comprehensive description of the experimental design and rigorous analysis of data. Overall, the functional readout data has tremendous value in clinical genetics (for helping resolving variants of uncertain significance) and computational genomics (for improving training or assessing of prediction methods). I recommend the manuscript to be accepted without revision. The following are discretionary minor comments:

1. Are SGE-enriched variants clustered in 3D? I understand that high-quality structure of DDX3X may not exist.
2. It's not very clear what are the features used for random forest training in the Methods section.

Point by point response to Reviewers

Reviewer #1, expertise in genomics, bioinformatics, systems biology and neuro disease (Remarks to the Author):

In this work, saturation genome editing (SGE) was performed on all 17 coding exons of DDX3X. Analysis of SGE results over time-course data revealed four categories of variants – unchanged, enriched, fast-depleting, and slow-depleting – showing the complex landscape of variant effects. To gain more insights into each group of variants, variant categories from SGE were compared to the functional annotations (e.g., missense), deleterious scoring (e.g., CADD), protein structure (e.g., buried, protein domain), effect on splicing, population cohort (e.g., GnomAD), and clinical annotations (e.g., ClinVar). A Random Forest Classifier was trained using SGE data to demonstrate the utility of SGE in distinguishing pathogenic variants from benign ones in neurodevelopmental disorders. Finally, the authors analyzed SGE behavior of DDX3X variants in cancer to explore potential role of DDX3X as a tumor suppressor gene.

Overall, the study is exciting and can benefit from addressing the following comments:

Several parts of manuscript and figures need improvement in clarity:

a. Figure 2:

We have now included a colour legend at the beginning of each figure, including Figure 2. The same colour key will apply to panels a-e, and we have updated the figure legend to reflect this.

ii. In panel c, what is the confidence interval? Is z-score calculated within each group or on all the data? Is the y-axis for each group directly comparable? The authors could consider normalizing each group to all start at 0 on Day 1.

Thank you for the suggestion. In order to normalize each group to all start at 0, we have changed the y-axis to cLFC (combined-LFC between both guides). Error bars denote the 95% Confidence Interval. The plot shown below will be included in the revised figure. The confidence intervals in some of the plots are not so visible simply because the error bars are so small.

Fig. R1

Response to Reviewers - Figure 1

cLFC over time for each of the SGE functional classification groups.

iii. Panels f and g, is the comparison with enriched group not significant or omitted here?

The enriched group is not significant. These plots have been removed, and replaced with Fig 3b and c in an expanded discussion of DDX3X protein structure.

b. Figure 5b, c: The 33 different cancer types are categorized into 4 groups (which seems to be overlapping groups?), do all cancer types in each group have the same % Enriched and % Depleted (x-axis)? How about instead showing a scatter plot with each point denoting each cancer type?

With the exception of medulloblastoma, most types of cancer have few *DDX3X* missense variants. Therefore the standard error is large when estimating the proportion of *DDX3X* missense that are driver variants for individual cancer types (Fig. R2).

Pooling the cancers where *DDX3X* appears to be a driver allows a more accurate and precise estimation of the proportion of missense variants that are drivers. However, we have provided the cancer-specific plots split by patient sex for all cancers with >1 missense *DDX3X* variant in Fig. S8. Plot shown below.

Fig. R2

Response to Reviewers - Figure 2: dNdScv analysis of individual cancer types

The proportion of missense variants classified as SGE-enriched and SGE-depleted (x-axis), and the estimated percentage of missense variants that are drivers (y-axis), in different cancer types where there is more than 1 missense *DDX3X* variant in A) females and B) males. Error bars denote 95% CI. CNS-MB: Central nervous system medulloblastoma; CESC: Cervical squamous cell carcinoma and endocervical adenocarcinoma; UCEC: Uterine Corpus Endometrial Carcinoma; HNSC: Head and Neck squamous cell carcinoma; KIRP : Kidney renal papillary cell carcinoma; KIRC : Kidney renal clear cell carcinoma; BLCA: Bladder Urothelial carcinoma; BRCA: Breast invasive carcinoma; Lymph: Lymphomas; OV: Ovarian serous cystadenocarcinoma.

c. Random forest classifier: What exactly are input features? Is input vector of length 1 showing the cLFC value of the respective variant or of length 3 with LFC on Day 7, 11 and 15?

The input vector is of length 3 comprising cLFC on Day 7, 11 and 15. We have now clarified this in the Methods.

2. It's great seeing random forest classifier trained using SGE data outperform classification using other deleterious scores (e.g., CADD) at clinically recommended thresholds. However, it is possible that the recommended threshold does not maximize for performance in this specific task. Could authors please also show performance comparison when training random forest classifier using CADD, REVEL, SIFT, and PolyPhen2 scores? Also has linear classifiers been tried to demonstrate the necessity of using a more complex nonlinear

model?

We appreciate that thresholds need to be task-specific, and so have reported the values at two commonly used thresholds (neither of which exceeds SGE data performance). However, we also report the ROC-AUC in Table 1 which is a metric of performance that is threshold-independent.

In response to your suggestion we have tested training the random forest classifier using both CADD scores and SGE cLFC at day 7, 11 and 15. This slightly reduced performance of the model compared to a classifier trained using only the cLFC values (sensitivity of 96.4% compared to 97.1%, specificity of 99.1% compared to 100%), see Table R1 below. However, given that only a single additional variant was mis-classified in the combined CADD+cLFC model, this difference in model performance should not be over-interpreted. This analysis has been included in Table S2. REVEL, SIFT, and PolyPhen2 scores are only available for missense variants and therefore were not used for training a random forest classifier for all variant consequence classes. In currently accepted clinical diagnostic interpretation guidelines (ACMG), evidence of variant effect from functional assays and *in silico* variant effect predictors must be scored and considered independently. Therefore, to enable diagnostic usage of our data, our classifier was intentionally derived only from the experimental SGE data and did not include *in silico*-derived information.

Table R1

Model	ROC AUC	TP	TN	FP	FN	Sens	Spec	PPV	NPV
RF	0.985	33	108	0	1	97.1	100	100	99.1
RF+ CADD	0.977	27	106	1	1	96.4	99.1	96.4	99.1

Response to Reviewers - Table 1. Performance comparison of Random Forest supervised classifier trained on cLFC at day 7, 11 and 15 with (RF+CADD) and without (RF) CADD scores.

ROC AUC = Area under the Receiver-operator characteristic (ROC) curve; TP = the number of test-positive true-positive variants. TN = the number of test-negative true-

negative variants. FP = the number of test-positive true-negative variants. FN = the number of test-negative true-positive variants. The number of true positive and true negative variants differs between the two models (RF and RF+CADD) as CADD scores are not available for all variants.

In response to your question regarding linear classifiers, three linear classifiers - logistic regression (LogReg), naive Bayes (NB) and linear discriminant analysis (LDA) classifiers - were tested (Table R2). The random forest (RF) model only marginally outperforms these linear classifiers, suggesting that the performance of the RF classifier is primarily driven by the SGE data, rather than the non-linearity of the model. These data have now been included as Table S2 and the text and methods updated.

Table R2

Model	ROC AUC	TP	TN	FP	FN	Sens	Spec	PPV	NPV
RF	0.985	33	108	0	1	97.1	100	100	99.1
NB	0.981	33	107	1	1	97.1	99.1	97.1	99.1
Log Reg	0.956	31	108	0	3	91.2	100	100	97.3
LDA	0.941	30	108	0	4	88.2	100	100	96.4

3. It is very interesting that two distinct modes of depletion (i.e., fast-depleting and slow-depleting) exist among variants. What are potential biological mechanisms causing this differentiation? Do fast and slow depleting variants cluster at different parts on 3D structure from PDB or AI predictions (e.g., AlphaFold or RoseTTAFold)? Could this be caused by a distinction in protein-protein interactions affected by different groups of variants?

We agree with the reviewer that this is an attractive hypothesis. SGE-depleting missense and codon-deletion variants are over-represented in the helicase domains (Fig. 3b,c), while enriched variants are more widely distributed through the protein. There is no evidence that fast-depleting missense variants have a more destabilising effect on protein structure than slow-depleting variants, as measured by delta-deltaG (Fig. 3a, Dunn's FDR=0.136). We have reassessed the distribution of fast and slow-

depleting variants within the three-dimensional structure of DDX3X by calculating the distance from the centroid of each amino-acid side-chain to the core of the protein. Analysis of both missense and codon-deletion variants suggests that fast-depleting variants tend to occur at residues closer to the protein core than slow-depleting variants (Figure R3 a-c, Supplementary Fig.S2. Dunn's FDR missense = 4.8×10^{-6} , codon deletion = 4.8×10^{-5}). However, both fast-depleting and slow-depleting variants are closer to the protein core than unchanged variants (Dunn's post-test missense slow-depleting FDR= 1.5×10^{-55} , fast-depleting FDR= 1.1×10^{-106} ; codon deletion slow-depleting FDR=0.02, fast-depleting FDR= 4.8×10^{-14}). This extends, and is consistent with, our previous analysis which used solvent accessible surface area (SASA), with a threshold of SASA < 25% indicating a buried residue. At this SASA threshold both fast and slow-depleting missense variants were significantly more likely to be buried.

We have also analysed the proportion of variants found at interaction interfaces with RNA, ATP and Mg²⁺ across the different SGE functional classes. SGE-depleting, particularly fast-depleting variants are significantly over-represented at interaction interfaces across both missense and codon deletion variants (fast-depleting variants missense χ^2 p= $3e^{-108}$, codon-deletion χ^2 p= $1.3e^{-4}$, Fig R3).

These plots have now been included in Fig. 3 and Supplementary Fig.S2. We have included supplementary pymol files showing the distribution of the different SGE functional classes on the three dimensional structure of DDX3X. Together these data support the hypothesis that slow-depleting variants may have a milder impact on DDX3X function than fast-depleting variants.

Fig. R3

Response to Reviewers - Figure 3: Properties of SGE-depleted and SGE-enriched variants

a) For each SGE functional class: Top panel – Observed/Expected number of *DDX3X* SNVs in UKBB and GnomAD, χ^2 test. Second panel: Amino acid conservation. Third panel: CADD PHRED scores. Fourth panel: $\Delta\Delta G$ for missense variants. Lower panel: distance from the centroid of *DDX3X* to the amino-acid side chain centroid (ångströms), missense variants. Dunn's FDR is shown in panels 2-5. b) The proportion of SGE functional classes in *DDX3X* missense variants stratified by their position in the protein. Interaction interface: residues in contact with RNA, magnesium ion or ATP. Buried residues: all residues with total solvent accessible surface area < 25%. χ^2 p-values relative to all missense are shown. c) The proportion of SGE functional classes in *DDX3X* codon-deletion variants stratified by their position in the protein. χ^2 p-values relative to all codon deletions are shown. d) AlphaFold2 *DDX3X* structure together with ATP, magnesium ion and RNA. Coloured according to the modal SGE functional class for missense variants at each residue. Spheres: residue main chain. Sticks: residue side chain.

Last but not the least, are there prognosis differences between fast-depleting and slow-depleting variants in cancer?

Thank you for this interesting suggestion. We have tried to test for differences in survival between individuals carrying fast-depleting and slow-depleting variants for CNS medulloblastoma, melanoma and diffuse large B-cell lymphoma. Unfortunately, due to a combination of sparse survival information (particularly in the medulloblastoma datasets) and modest variant numbers for each cancer type (medulloblastoma: 23 fast-depleting variants, 12 slow-depleting variants; melanoma: 3 fast-depleting variants, 5 slow-depleting variants; DLBCL: 8 fast-depleting variants, 6 slow-depleting variants) such an analysis was not possible.

Reviewer #2, expertise in genome editing, functional genomics, variant functional characterisation and cancer genomics (Remarks to the Author):

DDX3X Review

Radford, Tan et al. present a herculean effort to generate functional scores for all possible SNVs and some deletions for DDX3X by saturation genome editing (SGE). This work is of great significance for multiple reasons 1) this is the largest and most comprehensive sequence - function map generated by SGE to date, 2) it solves most of the variant classification problem for an important gene underlying developmental delay in females, 3) the authors validate the results and translate them into evidence for immediate use in clinical variant interpretation workflows. The data are highly sensitive and specific for accurately predicting clinical outcomes. However, there are a few analyses, explanations and figure additions/modifications (listed below) that we believe may strengthen the results as presented and add another level of transparency for readers and consumers of this data. We believe this work should be accepted and published as soon as possible.

MAJOR COMMENTS

1. Using DESeq2 might make sense for this type of read count data, but is not a precedent for MAVE. DESeq2 is used for determining differentially expressed genes or chromatin peaks where one does not know the truth. Here, that is not the case, for an assay measuring LoF, most stop codons and most synonymous codons can be used as an internal truth set. Same for known pathogenic and benign variants. Both of these types of “truth” variants can be used to derive sensible thresholds for functional categories. Can you explain why DESeq2 is a good choice for this dataset?

Thank you for raising the concern about using DESeq2 in our analysis. We agree that the use of “truth” variants allows for the empirical derivation of thresholds in SGE datasets, and we have done so in our analysis. We used DESeq2 to model dispersion, similar to other methods available for read-count quantification such as MAGeCK (<https://doi.org/10.1186/s13059-014-0554-4>). As described in the DESeq2 paper (<https://doi.org/10.1186/s13059-014-0550-8>), the dispersion estimation model was designed for experiments with small sample size/replicate number, which is generally true for MAVE or other biological experiments. However, we did not apply the DESeq2 default settings, where DESeq2 defines the log fold-change (LFC) baseline by assuming that most events have no significant change after the experimental treatment. As the reviewer correctly points out, this is not necessarily true for a MAVE dataset, and the majority of events (variants) in an exon may deplete if the exon encodes, for

example, an important protein domain. Therefore, we agree with the Reviewer that in a MAVE dataset, one should define a set of variants known to have no functional impact as the internal truth set for normalizing the LFC baseline. In order to do this with DESeq2, we did the following:

1. Replace the default scaling factor (by median ratio normalization) in DESeq2 with the total count normalization factor. By doing this, DESeq2 no longer assumes most of the tested events have no significant change.
2. Output the unshrunk/raw LFC from the DESeq2 package. By default, the DESeq2 shrinks the calculated LFC. The priors used for the shrinkage were designed for RNA-seq and, therefore, we disabled this in our analysis.
3. Synonymous and intronic variants not within the splicing regions (defined by VEP) were used as the internal truth set for LFC normalisation. We first calculate the median LFC of this variant set and subtract this value from the calculated LFC from step 2 described above. To ensure we have enough synonymous variants for calculating the median robustly, we included all possible synonymous variants for each codon in the SGE design.

We have now modified the text in the Methods section to more clearly highlight the detailed changes from the DESeq2 default settings. The text now reads “To use the synonymous and intronic variants as an internal control, the median of the raw LFC/LFC-trend from the sequences annotated as “synonymous_variant” and “intronic_variant” by VEP were subtracted from the LFC/LFC-trend of each variant.”

In summary, for this analysis, we have only used the dispersion estimation model and the negative binomial generalized linear model built into the DESeq2 package.

The following MA plots illustrate the impact of different normalization strategies for identifying significant variants. The default setting of DESeq2 tends to identify more enriched variants in exons with a large proportion of depleting variants. Normalizing the LFC with negative control variants (synonymous and intronic) can mitigate this artifact.

Figure R4

Response to Reviewers - Figure 4.

MA-plots showing the results of using different normalization strategies with the Day7 LFC for *DDX3X* Exon 14 sg2. Left plot: LFC normalised against synonymous and intronic negative control variants. Right plot: default DESeq2 analysis.

A. As deployed there is no intermediate function category, why was that decision made?

On reflection, we have now added an intermediate function category for the output of the random forest classifier. We believe that an intermediate function category has most utility in the clinical application of this model. While we have assessed the performance of this model using a posterior-probability threshold of 50%, we accept that in some clinical decision-making contexts a higher posterior-probability may be desirable.

Therefore, we propose that for clinical use a posterior probability of > 90% be considered high confidence, and posterior probability 50-90% be considered intermediate confidence (see Fig. R5 below, now included as a new supplemental figure Fig S7). We provide this new classification in an updated version of Table S6, for use in clinical diagnostic workflows.

Figure R5

Response to Reviewers - Figure 5

A) Posterior probability that a variant is non-functional from the Random Forest classifier (y-axis) plotted against the Day 15 cLFC. Fast and slow-depleting variants cluster above a posterior probability threshold of 0.9 (grey dotted line). B) Distribution of variants' Random Forest classifier posterior probability. Grey dotted line posterior probability = 0.9

B. What do the known pathogenic and benign variants look like when mapped onto Fig 2a or Fig 2b?

We have included the plots below in Fig. S2.

Fig. R6

C. What do synonymous and nonsense variants look like when mapped onto Fig 2a or Fig 2b?

We have included the plots below in Fig. 2 and Fig. S2, respectively.

Fig. R7

2. There are 6 measurements for each variant (2 guides X 3 technical replicates). What does the distribution error between those measurements look like? Do some target regions have higher error than others? What correlates with the error (e.g. read count? specific targets? gDNA quant?)? Can the error be taken into account for the clinical validation (e.g. are you equally confident in each variant score or RF classification or should some be down weighted or tossed entirely)?

The cLFC is calculated using an inverse-variance weighted average of the LFC of the two guide RNAs, and therefore adjusts for noise between the two guides. As the Random Forest classifier is trained using the cLFC at days 7, 11, 15 it incorporates some error adjustment. Standard error (SE) is highest for variants that have low read counts (Fig R8A, B, C). The predominant cause for low read counts is the degree to which a variant is depleted from the cellular population. Therefore, loss-of-function (LOF) variants tend to have a higher SE. However, LOF variants are also those where we have the highest confidence of clinical effect, and therefore SE is not likely to be a useful means of determining our confidence level in their classification. Exons which have a large number of SGE-depleting variants therefore have a higher error than exons with very few depleting variants (Fig R8D). However, we agree that not all Random Forest classifications are likely to be equivalent. One possibility would be to include the error term in the RF training dataset. This would be justified if we observed that the model had erroneously classified benign variants as nonfunctional. However, the model has perfect specificity, suggesting that this is not required. Therefore, we

propose to use the posterior-probability of the classifier to delineate an intermediate confidence group of variants, as described in section R2.B above.

Calculation of standard error: for each sgRNA, we have taken 3 measurements for each of the 5 timepoints for each variant. The dispersion of each variant was estimated by DESeq2, from the 15 measurements of each sgRNA. The standard error of the LFC-trend was derived from the negative binomial generalized linear model together with this estimated dispersion. The combined standard error of the cLFC-trend was calculated from the standard error from the sg1 dataset and sg2 dataset.

Fig. R8

Response to Reviewers - Figure 8: cLFC-trend standard error is negatively correlated with variants' mean read count.

A) Log_2 mean read count (y-axis) vs cLFC-trend standard error (x axis). B) Error distribution (scatter plot) for the different classes of variant. C) cLFC standard error for each exon.

3. Another addition to fig 2 (or the supplement) would be a plot of LfC by MAF. Evolution has already done the experiment. It would be good to see the correlation by functional score and minor allele frequency from GnomAD/UKBB.

While we agree that this is intuitively interesting, in practice as almost all variants observed in GnomAD and UKBB are unchanged with modest cLFCs these plots are relatively uninformative (Fig. R9).

Fig. R9

Response to Reviewers - Figure 9: Correlation of minor allele frequency and LFC i) Day 15 cLFC (y-axis) and minor allele frequency (MAF) (x-axis), coloured by SGE functional classification ii) violinplots of MAF for each of the observed SGE functional classification groups U = unchanged, E = enriched SD = slow-depleting for A) UK Biobank and B) GnomAD datasets.

4. There is an enriched population of variants according to the rate of growth in Fig 2c. It seems as though these variants are enriched in exons 3, 6, 7. However, these exons are outside of the folded functional domains and don't seem to be important for protein function given the missense and deletion data. Can your team speculate on a proposed mechanism?

The enriched variants are not clustered within the helicase domains, interaction interfaces or the protein core, unlike depleted variants (Fig. R3, Fig. 3), but are more distributed throughout the protein (Fig.3, Fig. S2e,f). Therefore, it is unlikely that they

are exerting an effect through altering specific biophysical interactions. It is possible that they result in enhanced translation of the protein, which confers a growth advantage to the cells. However, we feel that these speculations are sufficiently uninformed to not merit their detailed discussion. We have expanded the discussion of variant functional classes and protein structure, as described above, in R1.D.

A. Like above, we think it would be beneficial to show the distribution of the LFC of the synonymous variants to show that the enriched variants are not technical noise.

We have added the below plot to Fig. S2. The fact that enriched variants are significantly depleted from population cohorts such as GnomAD and UKBB, and exhibit different evolutionary and biophysical properties to unchanged variants, also suggests that these variants are not technical artefacts.

Fig. R10

Response to reviewers - Figure 10: Day 15 LFC distribution of synonymous and enriched variants.

5. Random forest model: it would be nice to map the random forest classes back onto either the scatter plot or histogram like fig 2a, b.

This has been added to Fig. 6.

Fig. R11

Response to Reviewers Figure - 11: A) Day 15 cLFC of variant abundance and B) Day 7 and Day 15 cLFC of variant abundance, coloured by the classification made by the Random Forest model with variants of NDD-relevance.

6. An HGVS string for the variant is missing from the final data set.

Thank you for identifying this. HGVSg, HGVS_p, and HGVS_c have now been added to the final data set.

7. The discussion ends with a call to arms for applying MAVE or SGE to more genes to generate functional evidence that can be used to interpret variants. Can you speculate on what that would take given what your team learned during the course of these experiments?

Two guides per target were used here. Is that necessary?

Can these experiments be scaled down?

Thank you for this suggestion, we have now added the following to the text:

“Our screen was performed with two sgRNAs per exon and with 5 timepoints. We have reanalyzed the data by dropping an sgRNA and multiple timepoints in the dataset to test whether we could simplify the experiment without affecting the sensitivity. We observe a ROC-AUC of 0.988 when using only 3 timepoints, and 0.97-0.99 when using only 1 sgRNA per exon. This suggests that SGE could potentially be scaled down to a single sgRNA per target with 3 timepoints without compromising data quality or clinical utility. “

What level of accuracy is needed for use in the clinic?

We are uncertain whether the reviewer is referring to the level of accuracy of the assay or predicting the function of individual variants.

With regards to assessing the level of accuracy of the assay, the 2019 guideline on the use of functional data in clinical diagnostic workflows calculates the odds of pathogenicity and weights the functional data accordingly (Brnich *et al.* 2019). This framework allows for robust data to be weighted as “strong” or “very strong” within clinical diagnostic workflows, but also allows for the conservative weighting of lower confidence functional data.

At the level of individual variants, we have used a Random Forest posterior-probability threshold of 50%. The corresponding sensitivity of over 97% and specificity of 100% suggest that this is a reasonable threshold for clinical use. However, we do recognise that there are some use-cases of these data, for example when considering *in utero* testing of variants, where clinicians may want greater confidence in the individual variant’s classification. For this reason we have introduced an intermediate confidence classification, detailed in Table S6, as described in R2.B above.

Is LoF in HAP1 an appropriate model for all DDD genes?

No, however SGE relies on the gene being essential in the cell line used. In addition, it can only test genes with pathophysiological loss-of-function effects. We have clarified this in the last paragraph of the discussion.

MINOR COMMENTS

- 1. The variant effect map is buried in Fig. S2a. The map in the supplement is new for the MAVE community, but that’s progress! Fig S2 isn’t terribly clear, however.**
 - A. Maybe make a separate map of the codon deletions (they are hard to see)?**
 - B. Maybe make it its own figure instead of smooshing it in with Fig S2b?**

Fig. S2 has been split into Fig. S2 and Fig. S3, with a separate map of the codon deletions.

C. We know we're biased, but we really like the sequence function map in Findlay et al (Figure 4). Since the sequence function map is in the supplement anyway, why not take up the space?

We like this map format too. The sequence function maps are now found in Fig. S3 and Fig. S6.

D. It would also be great to include the exon and functional motif cartoons in the main text figure so readers (reviewers) don't have to flip back and forth.

We have included the exon and functional motif cartoons in the supplementary figures as well as the main text figures.

2. There are 3 different protocols for tissue culture on your github

Thank you for identifying this, we have corrected this.

3. We think it would be beneficial to map the variant effects onto the protein structure to augment the genetic data. We feel like a more robust discussion of the effects of variation on protein structure/function could be added.

Thank you, this has been added, please see the response to reviewer 1 above.

4. More analysis of variant effect predictors. Are there predictors that do better or worse on different parts of the protein?

We have compared the performance of variant effect predictors across the protein. All *in silico* tools lack specificity, but Revel performs best, as previously reported. These plots are shown in Fig. 6 d,e.

Fig. R12

Response to Reviewers Figure - 12: Comparison of SGE data with that of *in silico* variant effect predictors, across the DDX3X protein. X-axis, variant position along the protein. A) all coding SNVs, B) Missense variants.

5. In page 9 paragraph 2, it is stated that in-frame exon 3 has a higher percentage of SGE-unchanged variants than other exons. Are you suggesting the protein is functional without exon 3? Has alternative splicing of DDX3X been identified in RNA-seq data?

GTEx data suggest the existence of two transcripts ENST00000457138.6 and ENST00000631641.1 which do not contain exon 3. However, these are expressed at very low levels, and are not expressed in the brain. To our knowledge, the function of the encoded isoforms has not been experimentally tested. This has been clarified in the text.

6. The discussion of the “wrong” SGE results for DDX3X variants in males. We’re not experts in DDD but it seems as though those are highly likely to be incorrect classifications. The self flagellation in the discussion is not warranted. (Any

chance y'all can take a second look at the genomes/exomes for those patients?)

Much of the clinical community is strongly in favour of variants in *DDX3X* causing a developmental disorder in males. Therefore we wanted to present a rigorous and reasonably comprehensive discussion of the possible explanations for the discordance with our results. We did not intend to be self flagellatory, and would welcome suggested changes to the text.

7. *DDX3Y* isn't expressed in HAP1 cells since HAP1 lacks the Y chromosome. Given *DDX3X* and *DDX3Y* share 92% homology, could you include some brief discussion on this model towards the study of *DDX3X*-related intellectual disability in males?

Loss of *DDX3X* in male mouse embryos results in early post-implantation lethality due to epiblast defects (doi: 10.1093/hmg/ddw143), suggesting that *DDX3Y* is not able to compensate for *DDX3X* in early development. *DDX3Y* is primarily expressed in the tissues of the male reproductive tract including the testes, seminal vesicles and prostate. However, a recent paper has suggested that in the mouse upregulation of *DDX3Y* could partially compensate for the loss of *DDX3X* in male brains after mid-gestation (DOI: [10.1016/j.devcel.2020.05.027](https://doi.org/10.1016/j.devcel.2020.05.027)). However, if this was the case during human development, we would expect to observe deleterious variants in healthy male individuals in GnomAD and UKBB, which we do not. It is theoretically possible that the presence of *DDX3Y* may potentiate the effect of *DDX3X* variants in male individuals. However, this is highly speculative and therefore we have not discussed this further in the manuscript.

Reviewer #3, expertise in machine learning, computational genomics, variant effect prediction and cancer genomics (Remarks to the Author):

The manuscript by Radford, Hurles, and colleagues describes multiplexed assays of variant effect (MAVS) of DDX3X, one of the most important risk genes of neurodevelopmental disorders. The in vitro functional assessment of DDX3X variants was done in HAP1 cells, a system used previously in MAVS of genes like BRCA1. About 3000 functionally abnormal variants were identified based on abundance at multiple time points. These variants can be grouped in 3 categories that may represent loss of function, hypo-morphic, and possibly gain of function. The manuscript has comprehensive description of the experimental design and rigorous analysis of data. Overall, the functional readout data has tremendous value in clinical genetics (for helping resolving variants of uncertain significance) and computational genomics (for improving training or assessing of prediction methods). I recommend the manuscript to be accepted without revision. The following are discretionary minor comments:

1. Are SGE-enriched variants clustered in 3D? I understand that high-quality structure of DDX3X may not exist.

Please see the expanded discussion of the SGE data and protein structure above, in response to reviewers 1 and 2.

2. It's not very clear what are the features used for random forest training in the Methods section.

cLFC at days 7, 11 and 15 were used. This has now been clarified in the Methods section.

REVIEWERS' COMMENTS

Reviewer #1 (Remarks to the Author):

The authors have thoroughly taken into account my previous remarks. This remarkable piece of research is bound to become a valuable asset for the community, and I wholeheartedly endorse its publication in its present state.

Reviewer #2 (Remarks to the Author):

This reviewer is satisfied with the responses of the authors.